# ALIGN-RUDDER: LEARNING FROM FEW DEMONSTRATIONS BY REWARD REDISTRIBUTION

## ABSTRACT

Reinforcement Learning algorithms require a large number of samples to solve complex tasks with sparse and delayed rewards. Complex tasks are often hierarchically composed of sub-tasks. Solving a sub-task increases the return expectation and leads to a step in the $Q$-function. RUDDER identifies these steps and then redistributes reward to them, thus immediately giving reward if sub-tasks are solved. Since the delay of rewards is reduced, learning is considerably sped up. However, for complex tasks, current exploration strategies struggle with discovering episodes with high rewards. Therefore, we assume that episodes with high rewards are given as demonstrations and do not have to be discovered by exploration. Unfortunately, the number of demonstrations is typically small and RUDDER's LSTM as a deep learning model does not learn well on these few training samples. Hence, we introduce Align-RUDDER, which is RUDDER with two major modifications. First, Align-RUDDER assumes that episodes with high rewards are given as demonstrations, replacing RUDDER's safe exploration and lessons replay buffer. Second, we substitute RUDDER's LSTM model by a profile model that is obtained from multiple sequence alignment of demonstrations. Profile models can be constructed from as few as two demonstrations. Align-RUDDER uses reward redistribution to speed up learning by reducing the delay of rewards. Align-RUDDER outperforms competitors on complex artificial tasks with delayed rewards and few demonstrations. On the MineCraft `ObtainDiamond` task, Align-RUDDER is able to mine a diamond, though not frequently.

## 1 INTRODUCTION

Reinforcement learning algorithms struggle with learning complex tasks that have sparse and delayed rewards (Sutton & Barto, 2018; Rahmandad et al., 2009; Luoma et al., 2017). For delayed rewards, temporal difference (TD) suffers from vanishing information (Arjona-Medina et al., 2019). On the other hand Monte Carlo (MC) has high variance since it must average over all possible futures (Arjona-Medina et al., 2019). Monte-Carlo Tree Search (MCTS), used for Go and chess, can handle delayed and rare rewards since it has a perfect environment model (Silver et al., 2016; 2017). RUDDER (Arjona-Medina et al., 2019; 2018) has been shown to excel in model-free learning of policies when only sparse and delayed rewards are given. RUDDER requires episodes with high rewards to store them in its lessons replay buffer for learning a reward redistribution model like an LSTM network. However, for complex tasks, current exploration strategies find episodes with high rewards only after an incommensurate long time. Humans and animals obtain high reward episodes by teachers, role models, or prototypes. Along this line, we assume that episodes with high rewards are given as demonstrations. Since generating demonstrations is often tedious for humans and time-consuming for exploration strategies, typically, only a few demonstrations are available. However, RUDDER's LSTM (Hochreiter, 1991; Hochreiter & Schmidhuber, 1997a) as a deep learning method requires many examples for learning. Therefore, we introduce Align-RUDDER, which replaces RUDDER's LSTM with a profile model obtained from multiple sequence alignment (MSA) of the demonstrations. Profile models are well known in bioinformatics. They are used to score new sequences according to their sequence similarity to the aligned sequences. Like RUDDER also Align-RUDDER performs reward redistribution —using an alignment model—, which considerably speeds up learning even if only a few demonstrations are available.

Figure 1: **Basic insight into reward redistribution. Left panel, Row 1**: An agent has to take a key to unlock a door. Both events increase the probability of receiving the treasure, which the agent always gets as delayed reward, when the door is unlocked at sequence end. **Row 2**: The $Q$-function approximation typically predicts the expected return at every state-action pair (red arrows). **Row 3**: However, the $Q$-function approximation requires only to predict the steps (red arrows). **Right panel, Row 1**: The $Q$-function is the future expected return (blue curve). Green arrows indicate $Q$-function steps and the big red arrow the delayed reward at sequence end. **Row 2 and 3**: The redistributed rewards correspond to steps in the $Q$-function (small red arrows). **Row 4**: After redistributing the reward, only the redistributed immediate reward remains (red arrows). Reward is no longer delayed.

Our main contributions are:

- We suggest a reinforcement algorithm that works well for sparse and delayed rewards, where standard exploration fails but few demonstrations with high rewards are available.

- We adopt multiple sequence alignment from bioinformatics to construct a reward redistribution technique that works with few demonstrations.

- We propose a method that uses alignment techniques and reward redistribution for identifying sub-goals and sub-tasks which in turn allow for hierarchical reinforcement learning.

## 2 REVIEW OF RUDDER

**Basic insight: $Q$-functions for complex tasks are step functions.** Complex tasks are typically composed of sub-tasks. Therefore the $Q$-function of an optimal policy resembles a step function. The $Q$-function is the expected future return and it increases (i.e, makes a step) when a sub-task is completed. Identifying large steps in the $Q$-function speeds up learning since it allows (i) to increase the return by performing actions that cause the step and (ii) to sample episodes with a larger return for learning.

An approximation to the $Q$-function must predict the expected future return for every state-action pair. However, a $Q$-function that resembles a step-function is mostly constant. Therefore predictions are only necessary at the steps. We have to identify the relevant state-actions that cause the steps and then predict the size of the steps. An LSTM network (Hochreiter, 1991; Hochreiter & Schmidhuber, 1995; 1997a;b) can identify relevant state-actions that open the input gate to store the size of the steps in the memory cells. Consequently, LSTM only updates its states and changes its return prediction when a new relevant state-action pair is observed. Therefore, both the change of the prediction and opening input gates indicate $Q$-function steps through an LSTM network that predicts the return of an episode.

**Reward Redistribution.** We consider episodic Markov decision processes (MDPs), i.e., the reward is only given once at the end of the sequence. The $Q$-function is assumed to be a step function, that is, the task can be decomposed into sub-tasks (see previous paragraph). *Reward redistribution* aims at giving the differences in the $Q$-function of an optimal policy as a new immediate reward. Since the $Q$-function of an optimal policy is not known, we approximate it by predicting the expected return by an LSTM network or by an alignment model in this work. The differences in predictions determine the reward redistribution. The prediction model will first identify the largest steps in the $Q$-function as they decrease the prediction error most. Fortunately, just identifying the largest steps even with poor predictions speeds up learning considerably. See Figure 1 for a description of the reward redistribution.

Figure 2: The function of a protein is largely determined by its structure. The relevant regions of this structure are even conserved across organisms, as shown in the left panel. Similarly, solving a task can often be decomposed into sub-tasks which are conserved across multiple demonstrations. As shown in the right panel where events are mapped to the letter code for amino acids. Sequence alignment makes those conserved regions visible and enables redistribution of reward to important events.

**Learning methods based on reward redistribution.** The redistributed reward serves as reward for a subsequent learning method: (A) The $Q$-values can be directly estimated (Arjona-Medina et al., 2019), which is used in the experiments for the artificial tasks and BC pre-training for MineCraft. (B) Redistributed rewards can serve for learning with policy gradients like Proximal Policy Optimization (PPO) (Schulman et al., 2018), which is used in the MineCraft experiments. (C) Redistributed rewards can serve for temporal difference learning like $Q$-learning (Watkins, 1989).

**LSTM models for reward redistribution.** RUDDER uses an LSTM model for predicting the future return. The reward redistribution is the difference between two subsequent predictions. If a state-action pair increases the prediction of the return, then it is immediately rewarded. Using state-action sub-sequences $(s, a)_{0:t} = (s_0, a_0, \ldots, s_t, a_t)$, the redistributed reward is $R_{t+1} = g((s, a)_{0:t}) - g((s, a)_{0:t-1})$, where $g$ is an LSTM model that predicts the return of the episode. The LSTM model learns at first to approximate the largest steps of the $Q$-function since they reduce the prediction error the most.

## 3 ALIGN-RUDDER: RUDDER WITH FEW DEMONSTRATIONS

In bioinformatics, sequence alignment identifies similarities between biological sequences to determine their evolutionary relationship (Needleman & Wunsch, 1970; Smith & Waterman, 1981). The result of the alignment of multiple sequences is a profile model. The profile model is a consensus sequence, a frequency matrix, or a Position-Specific Scoring Matrix (PSSM) (Stormo et al., 1982). New sequences can be aligned to a profile model and receive an alignment score that indicates how well the new sequences agree to the profile model.

Align-RUDDER uses such alignment techniques to align two or more high return demonstrations. For the alignment, we assume that the demonstrations follow the same underlying strategy, therefore they are similar to each other analog to being evolutionary related. If the agent generates a state-action sequence $(s, a)_{0:t-1}$, then this sequence is aligned to the profile model $g$ giving a score $g((s, a)_{0:t-1})$. The next action of the agent extends the state-action sequence by one state-action pair $(s_t, a_t)$. The extended sequence $(s, a)_{0:t}$ is also aligned to the profile model $g$ giving another score $g((s, a)_{0:t})$. The redistributed reward $R_{t+1}$ is the difference of these scores: $R_{t+1} = g((s, a)_{0:t}) - g((s, a)_{0:t-1})$ (see Eq. (1)). This difference indicates how much of the return is gained or lost by a adding another sequence element.

Align-RUDDER scores how close an agent follows an underlying strategy, which has been extracted by the profile model. Similar to the LSTM model, we identify the largest steps in the $Q$-function via relevant events determined by the profile model. Therefore, redistributing the reward by sequence alignment fits into the RUDDER framework with all its theoretical guarantees. RUDDER's theory for reward redistribution is valid for LSTM, other recurrent networks, attention mechanisms, or sequence and profile models.

**Advantages of alignment compared to LSTM.** Learning an LSTM model is severely limited when very few demonstrations are available. First, LSTM is known to require a large number of samples to generalize to new sequences. In contrast, sequence alignment requires only two examples to generalize well as known from bioinformatics. Second, expert demonstrations have high rewards.

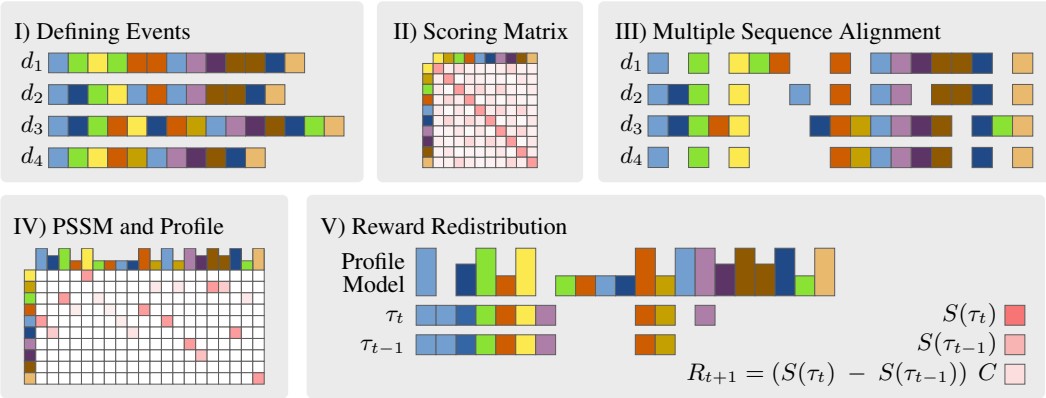

Figure 3: The five steps of Align-RUDDER's reward redistribution. **(I)** Define events and turn demonstrations into sequences of events. Each block represent an event to which the original state is mapped. **(II)** Construct a scoring matrix using event probabilities from demonstrations for diagonal elements and setting off-diagonal to a constant value. **(III)** Perform an MSA of the demonstrations. **(IV)** Compute a PSSM. Events with highest column scores are indicated at the top row. **(V)** Redistribute reward as the difference of scores of sub-sequences aligned to the profile.

Therefore random demonstrations with very low rewards have to be generated. LSTM does not generalize well when only these extreme reward cases can be observed in the training set. In contrast, sequence alignment only uses examples that are closely related; that is, they belong to the same category (expert demonstrations).

**Reward Redistribution by Sequence Alignment.** The new reward redistribution approach consists of five steps, see Fig. 3: (I) Define events to turn episodes of state-action sequences into sequences of events. (II) Determine an alignment scoring scheme, so that relevant events are aligned to each other. (III) Perform a multiple sequence alignment (MSA) of the demonstrations. (IV) Compute the profile model like a PSSM. (V) Redistribute the reward: Each sub-sequence $\tau_t$ of a new episode $\tau$ is aligned to the profile. The redistributed reward $R_{t+1}$ is proportional to the difference of scores $S$ based on the PSSM given in step (IV), i.e. $R_{t+1} \propto S(\tau_t) - S(\tau_{t-1})$.

In the following, the five steps of Align-RUDDER's reward redistribution are outlined. For the interested reader, each step is detailed in Sec. A.3 in the appendix. Finally, in Sec. A.7.3 in the appendix, we illustrate these five steps on the example of Minecraft.

**(I) Defining Events.** Instead of states, we consider differences of consecutive states to detect a change caused by an important event like achieving a sub-goal. An *event* is defined as a cluster of state differences. We use similarity-based clustering like affinity propagation (AP) (Frey & Dueck, 2007). If states are only enumerated, we suggest to use the "successor representation" (Dayan, 1993) or "successor features" (Barreto et al., 2017). We use the demonstrations combined with state-action sequences generated by a random policy to construct the successor representation.

A sequence of events is obtained from a state-action sequence by mapping states $s$ to its cluster identifier $e$ (the event) and ignoring the actions. Alignment techniques from bioinformatics assume sequences composed of a few events, e.g. 20 events. If there are too many events, good fitting alignments cannot be distinguished from random alignments. This effect is known in bioinformatics as "Inconsistency of Maximum Parsimony" (Felsenstein, 1978).

**(II) Determining the Alignment Scoring System.** A scoring matrix $\mathbb{S}$ with entries $\mathbb{s}_{i,j}$ determines the score for aligning event $i$ with $j$. A priori, we only know that a relevant event should be aligned to itself but not to other events. Therefore, we set $\mathbb{s}_{i,j} = 1/p_i$ for $i = j$ and $\mathbb{s}_{i,j} = \alpha$ for $i \neq j$. Here, $p_i$ is the relative frequency of event $i$ in the demonstrations. $\alpha$ is a hyper-parameter, which is typically a small negative number. This scoring scheme encourages alignment of rare events, for which $p_i$ is small. For more details see Appendix Sec. A.3.

**(III) Multiple sequence alignment (MSA).** An MSA algorithm maximizes the sum of all pairwise scores $S_{\text{MSA}} = \sum_{i,j,i<j} \sum_{t=0}^{L} \mathbb{s}_{i,j,t_i,t_j,t}$ in an alignment, where $\mathbb{s}_{i,j,t_i,t_j,t}$ is the score at alignment

column $t$ for aligning the event at position $t_i$ in sequence $i$ to the event at position $t_j$ in sequence $j$. $L \geq T$ is the alignment length, since gaps make the alignment longer than the length of each sequence. We use ClustalW (Thompson et al., 1994) for MSA. MSA constructs a guiding tree by agglomerative hierarchical clustering of pairwise alignments between all demonstrations. This guiding tree allows to identify multiple strategies. For more details see Appendix Sec. A.3.

**(IV) Position-Specific Scoring Matrix (PSSM) and MSA profile model.** From the alignment, we construct a profile model as a) column-wise event probabilities and b) a PSSM (Stormo et al., 1982). The PSSM is a column-wise scoring matrix to align new sequences to the profile model. More details are given in Appendix Sec. A.3.

**(V) Reward Redistribution.** The reward redistribution is based on the profile model. A sequence $\tau = e_{0:T}$ ($e_t$ is event at position $t$) is aligned to the profile, which gives the score $S(\tau) = \sum_{l=0}^{L} \mathbb{s}_{l,t_l}$. Here, $\mathbb{s}_{l,t_l}$ is the alignment score for event $e_{t_l}$ at position $l$ in the alignment. Alignment gaps are columns to which no event was aligned, which have $t_l = T + 1$ with gap penalty $\mathbb{s}_{l,T+1}$. If $\tau_t = e_{0:t}$ is the prefix sequence of $\tau$ of length $t + 1$, then the reward redistribution $R_{t+1}$ for $0 \leqslant t \leqslant T$ is

$$R_{t+1} = (S(\tau_t) - S(\tau_{t-1}))\ C = g((s,a)_{0:t}) - g((s,a)_{0:t-1}),\ R_{T+2} = \tilde{G}_0 - \sum_{t=0}^{T} R_{t+1}, \quad (1)$$

where $C = \mathrm{E}_{\text{demo}}\left[\tilde{G}_0\right] / \mathrm{E}_{\text{demo}}\left[\sum_{t=0}^{T} S(\tau_t) - S(\tau_{t-1})\right]$ with $S(\tau_{-1}) = 0$. The original return of the sequence $\tau$ is $\tilde{G}_0 = \sum_{t=0}^{T} \tilde{R}_{t+1}$ and the expectation of the return over demonstrations is $\mathrm{E}_{\text{demo}}$. The constant $C$ scales $R_{t+1}$ to the range of $\tilde{G}_0$. $R_{T+2}$ is the correction of the redistributed reward (Arjona-Medina et al., 2019), with zero expectation for demonstrations: $\mathrm{E}_{\text{demo}}[R_{T+2}] = 0$. Since $\tau_t = e_{0:t}$ and $e_t = f(s_t, a_t)$, we can set $g((s,a)_{0:t}) = S(\tau_t)C$. We ensure strict return equivalence (Arjona-Medina et al., 2019) by $G_0 = \sum_{t=0}^{T+1} R_{t+1} = \tilde{G}_0$. The redistributed reward depends only on the past: $R_{t+1} = h((s,a)_{0:t})$.

**Sub-tasks.** The reward redistribution identifies sub-tasks as alignment positions with high redistributed rewards. These sub-tasks are indicated by high scores $\mathbb{s}$ in the PSSM. Reward redistribution also determines the terminal states of sub-tasks since it assigns rewards for solving the sub-tasks. However, reward redistribution and Align-RUDDER cannot guarantee that the redistributed reward is Markov. For redistributed Markov reward, options (Sutton et al., 1999), MAXQ (Dietterich, 2000), or recursive option composition (Silver & Ciosek, 2012) can be used.

**Higher Order Markov Reward Redistributions.** Align-RUDDER may lead to higher-order Markov redistribution. Corollary 1 in the appendix states that the optimality criterion from Theorem 2 in Arjona-Medina et al. (2019) also holds for higher-order Markov reward redistributions. If the expected redistributed higher-order Markov reward is the difference of $Q$-values. In that case the redistribution is optimal, and there is no delayed reward. Furthermore, the optimal policies are the same as for the original problem. This corollary is the motivation for redistributing the reward to the steps in the $Q$-function. In the Appendix, Corollary 2 states that under a condition, an optimal higher-order reward redistribution can be expressed as the difference of $Q$-values.

## 4 EXPERIMENTS

Align-RUDDER is compared on three artificial tasks with sparse & delayed rewards and few demonstrations to Behavioral Cloning with $Q$-learning (BC+$Q$), Soft $Q$ Imitation Learning (SQIL) (Reddy et al., 2020), RUDDER (LSTM), and Deep $Q$-learning from Demonstrations (DQfD) (Hester et al., 2018). GAIL (Ho & Ermon, 2016) failed to solve the two artificial tasks, as reported previously for similar tasks (Reddy et al., 2020). Then, we test Align-RUDDER on the complex MineCraft `ObtainDiamond` task with episodic rewards (Guss et al., 2019b). All experiments use finite time MDPs with $\gamma = 1$ and episodic reward. More details are in Appendix Sec. A.6.

**Alignment vs LSTM in 1D key-chest environment.** We use a *1D key-chest environment* to show the effectiveness of sequence alignment in a low data regime compared to an LSTM model. The agent has to collect the key and then open the chest to get a positive reward at the last timestep. See Appendix Fig. A.9 for a schematic representation of the environment. As the key-events (important state-action pairs) in this environment are known, we can compute the *key-event detection rate* of

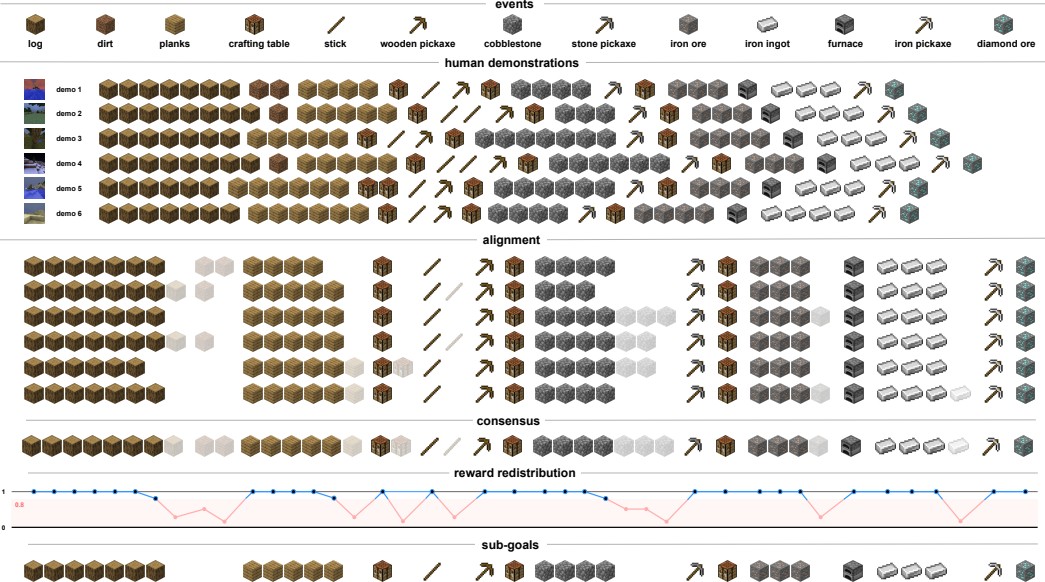

Figure 4: Example of alignment and reward redistribution for demonstrations of `ObtainDiamond`. Thresholding the redistributed reward identifies sub-goals.

a reward redistribution model. A key event is detected if the redistributed reward of an important state-action pair is larger than the average redistributed reward in the sequence. We train the reward redistribution models with 2, 5, and 10 training episodes and test on 1000 test episodes, averaged over ten trials. Align-RUDDER significantly outperforms LSTM (RUDDER) for detecting these key events in all cases, with an average key-event detection rate of 0.96 for sequence alignment vs. 0.46 for the LSTM models overall dataset sizes. See Appendix Fig. A.10 for the detailed results.

**Artificial tasks (I) and (II).** They are variations of the gridworld *rooms example* (Sutton et al., 1999), where cells are the MDP states. In our setting, the states do not have to be time-aware for ensuring stationary optimal policies, but the unobserved used-up time introduces a random effect. The grid is divided into rooms. The agent's goal is to reach a target from an initial state with the fewest steps. It has to cross different rooms, which are connected by doors, except for the first room, which is only connected to the second room by a *teleportation portal*. The portal is introduced to avoid BC initialization alone, solving the task. It enforces that going to the portal entry cells is learned when they are at positions not observed in demonstrations. At every location, the agent can move *up, down, left, right*. The state transitions are stochastic. An episode ends after $T = 200$ time steps. Suppose the agent arrives at the target. In that case, it goes into an absorbing state where it stays until $T = 200$ without receiving further rewards. The reward is only given at the end of the episode. Demonstrations are generated by an optimal policy with a 0.2 exploration rate.

The five steps of Align-RUDDER's reward redistribution are: (1) Events are clusters of states obtained by Affinity Propagation using as similarity the successor representation based on demonstrations. (2) The scoring matrix is obtained according to (II), using $\epsilon = 0$ and setting all off-diagonal values of the scoring matrix to $-1$. (3) ClustalW is used for the MSA of the demonstrations with zero gap penalties and no biological options. (4) The MSA supplies a profile model and a PSSM as in (IV). (5) Sequences generated by the agent are mapped to sequences of events according to (I). The reward is redistributed via differences of profile alignment scores of consecutive sub-sequences according to Eq. (1) using the PSSM. The reward redistribution determines sub-tasks like doors or portal arrival. The sub-tasks partition the $Q$-table into sub-tables that represent a sub-agent. However, we optimize a single $Q$-table in these experiments. Defining sub-tasks has no effect on learning in the tabular case.

All compared methods learn a $Q$-table and use an $\epsilon$-greedy policy with $\epsilon = 0.2$. The $Q$-table is initialized by behavioral cloning (BC). The state-action pairs which are not initialized since they are not visited in the demonstrations get an initialization by drawing a sample from a normal distribution. Align-RUDDER learns the $Q$-table via RUDDER's $Q$-value estimation (learning method (A) from

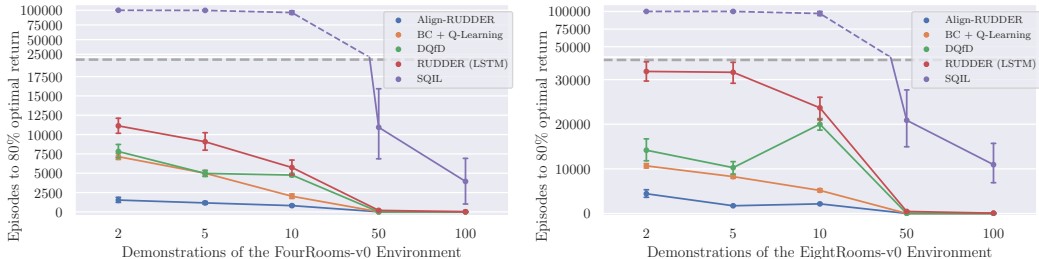

Figure 5: Comparison of Align-RUDDER and other methods on Task (I) (left) and Task (II) (right) with respect to the number of episodes required for learning on different numbers of demonstrations. Results are the average over 100 trials. Align-RUDDER significantly outperforms all other methods.

Sec. 2). For BC+$Q$, RUDDER (LSTM), SQIL, and DQfD a $Q$-table is learned by $Q$-learning. Hyperparameters are selected via grid search using the same amount of time for each method. For different numbers of demonstrations, performance is measured by the number of episodes to achieve 80% of the average return of the demonstrations. A Wilcoxon rank-sum test determines the significance of performance differences between Align-RUDDER and the other methods.

**Task (I)** environment is a $12 \times 12$ gridworld with four rooms. The target is in room #4, and the start is in room #1 with 20 portal entry locations. The state contains the portal entry for each episode. Fig. 5 shows the number of episodes required for achieving 80% of the average reward of the demonstrations for different numbers of demonstrations. Results are averaged over 100 trials. **Align-RUDDER significantly outperforms all other methods, for $\leqslant 10$ demonstrations ($p$-values $< 10^{-10}$).**
**Task (II)** is a $12 \times 24$ gridworld with eight rooms: target in room #8, and start in room #1 with 20 portal entry locations. Fig. 5 shows the results with settings as in Task (I). **Align-RUDDER significantly outperforms all other methods, for $\leqslant 10$ demonstrations ($p$-values $< 10^{-19}$).** We also conduct an ablation study to study performance of Align-RUDDER, while changing various parameters, like environment stochasticity (See Sec. A.6.4) and number of clusters (See Sec. A.6.5).

**MineCraft.** We further test Align-RUDDER on MineCraft ObtainDiamond task from the MineRL dataset (Guss et al., 2019b). We do not use intermediate rewards given by achieving sub-goals from the challenge, since Align-RUDDER is supposed to discover such sub-goals automatically via reward redistribution. We only give a reward for mining the diamond. This requires resource gathering and tool building in a hierarchical way. To the best of our knowledge, no pure learning method (sub-goals are also learned) has mined a diamond yet (Scheller et al., 2020). The dataset contains demonstrations which are insufficient to directly learn a single policy (117 demonstrations, 67 mined a diamond).

Implementation: (1) A state consists of visual input and an inventory (incl. equip state). Both inputs are normalized to the same information, that is, the same number of components and the same variance. We cluster the differences of consecutive states (Arjona-Medina et al., 2019). Very large clusters are removed, and small merged, giving 19 clusters corresponding to events, which are characterized by inventory changes. Finally, demonstrations are mapped to sequences of events. (2) The scoring matrix is computed according to (II). (3) The ten shortest demonstrations that obtained a diamond are aligned by ClustalW with zero gap penalties and no biological options. (4) The multiple alignments gives a profile model and a PSSM. (5) The reward is redistributed via differences of profile alignment scores of consecutive sub-sequences according to Eq. (1) using the PSSM. Based on the reward redistribution, we define sub-goals. Sub-goals are identified as profile model positions that obtain an average redistributed reward above a threshold for the demonstrations. Demonstration sub-sequences between sub-goals are considered as demonstrations for the sub-tasks. New sub-sequences generated by the agent are aligned to the profile model to determine whether a sub-goal is achieved. The redistributed reward between two sub-goals is given at the end of the sub-sequence, therefore, the sub-tasks also have an episodic reward. Fig. 4 shows how sub-goals are identified. Sub-agents are pre-trained on the demonstrations for the sub-tasks using BC, and further trained in the environment using Proximal Policy Optimization (PPO) (Schulman et al., 2018). BC pre-training corresponds to RUDDER's $Q$-value estimation (learning method (A) from above), while PPO corresponds to RUDDER's PPO training (learning method (B) from above).

Table 1: Maximum item score of methods on the MineCraft task. "Auto": Sub-goals/sub-tasks are found automatically. Demonstrations are used for hierarchical reinforcement learning ("HRL"). Methods: Soft-Actor Critic (SAC, Haarnoja et al. (2018)), DQfD, Meta Learning Shared Hierarchies (MLSH, Frans et al. (2018)), Rainbow (Hessel et al., 2017), PPO, and BC.

| Method | Team Name | HRL/Auto | | | | | | | | | |
|--------|-----------|----------|---|---|---|---|---|---|---|---|---|
| Align-RUDDER | Ours | ✓/✓ | | | | | | | | | |
| DQfD | CDS | ✓/✗ | | | | | | | | | |
| BC | MC_RL | ✓/— | | | | | | | | | |
| CLEAR | I4DS | ✗/✓ | | | | | | | | | |
| Options&PPO | CraftRL | ✓/✗ | | | | | | | | | |
| BC | UEFDRL | ✗/✓ | | | | | | | | | |
| SAC | TD240 | ✗/✓ | | | | | | | | | |
| MLSH | LAIR | ✓/✓ | | | | | | | | | |
| Rainbow | Elytra | ✗/✓ | | | | | | | | | |
| PPO | karolisram | ✗/✓ | | | | | | | | | |

Our main agent can perform all actions but additionally can execute sub-agents and learns via the redistributed reward. The main agent corresponds to and is treated like a Manager module (Vezhnevets et al., 2017). The main agent is initialized by executing sub-agents according to the alignment but can deviate from this strategy. When a sub-agent successfully completes its task, the main agent executes the next sub-agent according to the alignment. More details can be found in Appendix Sec. A.7.1. Using only ten demonstrations, Align-RUDDER is able to learn to mine a diamond. A diamond is obtained in **0.1%** of the cases. With 0.5 success probability for each of the 31 extracted sub-tasks (skilled agents not random agents), the resulting success rate for mining the diamond would be $4.66 \times 10^{-10}$. Tab. 1 shows a comparison of methods on the MineCraft MineRL dataset by the maximum item score (Milani et al., 2020). Results are taken from (Milani et al., 2020), in particular from Figure 2, and completed by (Skrynnik et al., 2019; Kanervisto et al., 2020; Scheller et al., 2020). Align-RUDDER was not evaluated during the MineCraft MineRL challenge, but it follows the timesteps limit (8 million) imposed by the challenge. Align-RUDDER did not receive the intermediate rewards provided by the challenge that hint at sub-tasks, thus tries to solve a more difficult task. Recently, ForgER++ (Skrynnik et al., 2020) was able to mine a diamond in **0.0667%** of the cases. We do not include it in Table 1 as it did not have any limitations on the number of timesteps. Also, ForgER++ generates sub-goals for MineCraft using a heuristic, while Align-RUDDER uses redistributed reward to automatically obtain sub-goals.

**Analysis of MineCraft Agent Behaviour.** For each agent and its sub-task, we estimate the success rate and its improvement during fine-tuning by averaging over return of multiple runs (see Fig. 6). For earlier sub-tasks, the agent has a relatively higher sub-task success rate. This also corresponds to the agent having access to much more data for earlier sub-tasks. During learning from demonstrations, much less data is available for training for later sub-tasks, as not all expert demonstrations achieve the later tasks. During online training using reinforcement learning, an agent has to successfully complete all earlier sub-tasks to generate trajectories for later sub-tasks. This is exponentially difficult. Lack of demonstrations and difficulty of the learned agent to generate data for later sub-tasks leads to degradation of the success rate in MineCraft.

## 5 RELATED WORK

Learning from demonstrations has been widely studied over the last 50 years (Billard et al., 2008). An example is imitation learning, which uses supervised techniques when the number of demonstrations is large enough (Michie et al., 1990; Pomerleau, 1991; Michie & Camacho, 1994; Schaal, 1996; Kakade & Langford, 2002). However, policies trained with imitation learning tend to drift away from demonstration trajectories due to a distribution shift (Ross & Bagnell, 2010). This effect can be mitigated (III et al., 2009; Ross & Bagnell, 2010; Ross et al., 2011; Judah et al., 2014; Sun et al., 2017;

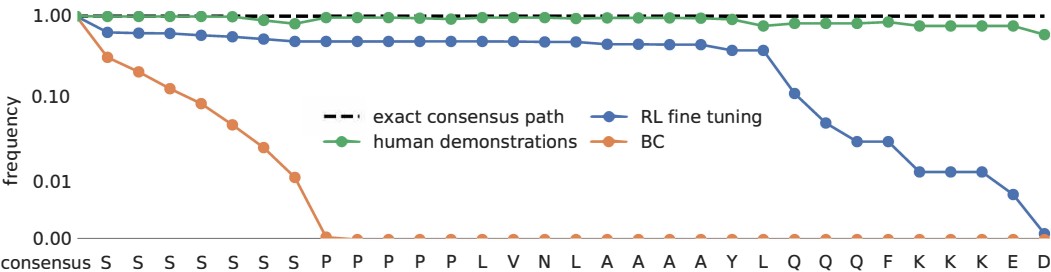

Figure 6: Comparing the consensus frequencies between behavioral cloning (BC, orange), where fine-tuning starts, the fine-tuned model (blue), and human demonstrations (green). The plot is in symmetric log scale (symlog in matplotlib). See Appendix Fig. A.19 for mapping of letters to items.

2018). Many approaches use demonstrations for initialization, e.g. of policy networks (Taylor et al., 2011; Silver et al., 2016), value function networks (Hester et al., 2017; 2018), both networks (Zhang & Ma, 2018; Nair et al., 2018), or an experience replay buffer (Hosu & Rebedea, 2016). Beyond initialization, demonstrations are used to define constraints (Kim et al., 2013), generate sub-goals (Eysenbach et al., 2019), enforce regularization (Reddy et al., 2020), guide exploration (Subramanian et al., 2016; Jing et al., 2019), or shape rewards (Judah et al., 2014; Brys et al., 2015; Suay et al., 2016). Demonstrations may serve for inverse reinforcement learning (Ng & Russell, 2000; Abbeel & Ng, 2004; Ho & Ermon, 2016), which aims at learning a (non-sparse) reward function that best explains the demonstrations. Learning reward functions requires a large number of demonstrations (Syed & Schapire, 2007; Ziebart et al., 2008; Silva et al., 2019). Some approaches rely on few-shot or/and meta learning (Duan et al., 2017; Finn et al., 2017; Zhou et al., 2020). However, few-shot and meta learning demand a large set of auxiliary tasks or prerecorded data. Concluding, most methods that learn from demonstrations rely on the availability of many demonstrations (Khardon, 1999; Lopes et al., 2009), in particular, if using deep learning methods (Bengio & Lecun, 2007; Lakshminarayanan et al., 2016). Some methods can learn on few demonstrations like Soft $Q$ Imitation Learning (SQIL) (Reddy et al., 2020), Generative Adversarial Imitation Learning (GAIL) (Ho & Ermon, 2016), and Deep $Q$-learning from Demonstrations (DQfD) (Hester et al., 2018).

## 6 DISCUSSION AND CONCLUSION

**Discussion.** Firstly, reward redistributions do not change the optimal policies (see Theorem 1 in Appendix). Thus, suboptimal reward redistributions due to alignment errors or choosing events that are non-essential for reaching the goal might not speed up learning, but also do not change the optimal policies. Secondly, while Align-RUDDER can speed up learning even in complex environments, the resulting performance depends on the quality of the alignment model. A low quality alignment model can arise from multiple factors, one of which is having large number ($\gg 20$) of distinct events. Clustering can be used to reduce the number of events, which could also lead to a low quality alignment model if too many relevant events are clustered together. While the optimal policy is not changed by poor demonstration alignment, the benefit of employing reward redistribution based on it diminishes. Thirdly, the alignment could fail if the demonstrations have different underlying strategies i.e no events are common in the demonstrations. We assume that the demonstrations follow the same underlying strategy, therefore they are similar to each other and can be aligned. However, if no underlying strategy exists, then identifying those relevant events via alignment, which should receive high redistributed rewards, may fail. In this case, reward is given at sequence end, when the redistributed reward is corrected, which leads to an episodic reward without reducing the delay of the rewards and speeding up learning.

**Conclusions.** We have introduced Align-RUDDER to solve highly complex tasks with delayed and sparse reward from few demonstrations. We have shown experimentally that Align-RUDDER outperforms state of the art methods designed for learning from demonstrations in the regime of few demonstrations. On the MineCraft `ObtainDiamond` task, Align-RUDDER is, to the best of our knowledge, the first pure learning method to mine a diamond.

ETHICS STATEMENT

**Impact on ML and related scientific fields.**   Our research has the potential to positively impact a wide variety of fields of life due to its general applicability. Most importantly, it has the potential to reduce the cost for training and deploying agents in real world applications and therefore enable systems that have not been possible until now.

However, any new development in machine learning can be applied for good or for bad. Our system can be used for medical applications where it can save life but it could be used for malevolent systems. It is the society that decides how new technology is employed. However, we as scientist have to inform the society and the decision makers about our technologies. We have to show the limits of our technology, to give ideas of possible applications, to point out possible misuse or erroneous operation of our new technology.

**Impact on society.**   A big danger is that users rely too much on our new approach and use it without reflecting on the outcomes. For example, in medical treatment decisions doctors may rely on the technical system and push are the responsibility toward the machine: "The machine suggested this treatment, therefore it is not my fault". Another example is self-driving cars where we see that drivers become more careless even if they are supposed to pay attention and keep the hands on the steering wheel. They trust too much in the technology, even if the technology does not justify this trust or is not mature.

Finally, our method can be deployed in companies for job automation. Therefore there is the danger that some people lose their jobs, particularly those whose work is to perform predictable and repetitive tasks. An often used example is the taxi driver who would lose their job because of self-driving cars. The same holds for many jobs in production industry where automation can replace jobs. However all industrialization led to loss of jobs but new jobs have been created.

**Consequences of failures of the method.**   Depending on the application area, a failure of this method might be of lesser concern, such as a failed execution of a computer program. If our method is employed within a larger automation system, a failure can result in damages such as a car accident. However, this holds for almost all reinforcement learning methods, and usage and testing falls within the responsibility of the application area. We note that in this work, the method was only used in computer game environments.

**Leveraging of biases in the data and potential discrimination.**   Our proposed method relies on human demonstrations and thereby human decisions, which are usually strongly biased. As almost all machine learning methods trained on human-influenced data, our method could learn to use and exploit those biases and make similar decisions (Solaiman et al., 2019). Therefore, the responsible use of our method depends on a careful selection of the training data and awareness of the potential biases within those.

REPRODUCIBILITY STATEMENT

Code for experiments on the FourRooms and EightRooms environment is included as supplementary material. The README contains step-by-step instructions to set up an environment and run the experiments. We have specified all the training details ex. hyperparameters and how they were chosen in the Appendix (See Section A.6). We trained 100 replicates for each datapoint of the first set of experiments and are shown in Fig. 5. Using the code in the supplementary material, it is quite easy to reproduce our results for these experiments.

We also include code for the experiments done for MineCraft in the supplementary materials. All the preprocessing steps, hyperparameters and other implementation details are given in the Appendix (See Section A.7).

We also provide a deeper overview of the RUDDER (Arjona-Medina et al., 2019) theory in the Appendix (See Section A.2) as it is important for many design choices in Align-RUDDER.

Finally, a video showcasing the MineCraft agent is also provided as supplementary material.

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

# A  APPENDIX

## CONTENTS OF THE APPENDIX

## LIST OF FIGURES

## A.1 Introduction to the Appendix

This is the appendix to the paper "Align-RUDDER: Learning from few Demonstrations by Reward Redistribution". The appendix aims at supporting the main document and provides more detailed information about the implementation of our method for different tasks. The content of this document is summarized as follows:

• Section A.3 describes the five steps of Align-RUDDER's reward redistribution in more detail. In particular, the scoring systems are described in more detail. • Section A.4 provides a brief overview of sequence alignment methods and the hyperparameters used in our experiments. • Section A.6 provides figures and tables to support the results of the experiments in Artificial Tasks (I) and (II). • Section A.7 explains in detail the experiments conducted in the Minecraft *ObtainDiamond* task.

## A.2 Review Reward Redistribution

Reward redistribution and return decomposition are concepts introduced in RUDDER but also apply to Align-RUDDER as it is a variant of RUDDER. Reward redistribution based on return decomposition eliminates – or at least mitigates – delays of rewards while preserving the same optimal policies. Align-RUDDER is justified by the theory of return decomposition and reward redistribution when using multiple sequence alignment for constructing a reward redistribution model. In this section, we review the concepts of return decomposition and reward redistribution.

**Preliminaries.** We consider a finite MDP defined by the 5-tuple $\mathcal{P} = (\mathcal{S}, \mathcal{A}, \mathcal{R}, p, \gamma)$ where the state space $\mathcal{S}$ and the action space $\mathcal{A}$ are sets of finite states $s$ and actions $a$ and $\mathcal{R}$ the set of bounded rewards $r$. For a given time step $t$, the corresponding random variables are $S_t$, $A_t$ and $R_{t+1}$. Furthermore, $\mathcal{P}$ has transition-reward distributions $p(S_{t+1} = s', R_{t+1} = r \mid S_t = s, A_t = a)$, and a discount factor $\gamma \in (0, 1]$, which we keep at $\gamma = 1$. A Markov policy $\pi(a \mid s)$ is a probability of an action $a$ given a state $s$. We consider MDPs with finite time horizon or with an absorbing state. The discounted return of a sequence of length $T$ at time $t$ is $G_t = \sum_{k=0}^{T-t} \gamma^k R_{t+k+1}$. As usual, the $Q$-function for a given policy $\pi$ is $q^\pi(s, a) = \mathrm{E}_\pi[G_t \mid S_t = s, A_t = a]$. $\mathrm{E}_\pi[x \mid s, a]$ is the expectation of $x$, where the random variable is a sequence of states, actions, and rewards that is generated with transition-reward distribution $p$, policy $\pi$, and starting at $(s, a)$. The goal is to find an optimal policy $\pi^* = \mathrm{argmax}_\pi \mathrm{E}_\pi[G_0]$ maximizing the expected return at $t = 0$. We assume that the states $s$ are time-aware (time $t$ can be extracted from each state) in order to assure stationary optimal policies. According to Proposition 4.4.3 in (Puterman, 2005), a deterministic optimal policy $\pi^*$ exists.

**Definitions.** A *sequence-Markov decision process* (SDP) is defined as a decision process that has Markov transition probabilities but a reward probability that is not required to be Markov. Two SDPs $\tilde{\mathcal{P}}$ and $\mathcal{P}$ with different reward probabilities are *return-equivalent* if they have the same expected return at $t = 0$ for each policy $\pi$, and *strictly return-equivalent* if they additionally have the same expected return for every episode. Since for every $\pi$ the expected return at $t = 0$ is the same, return-equivalent SDPs have the same optimal policies. A *reward redistribution* is a procedure that —for a given sequence of a delayed reward SDP $\tilde{\mathcal{P}}$— redistributes the realization or expectation of its return $\tilde{G}_0$ along the sequence. This yields a new SDP $\mathcal{P}$ with $R$ as random variable for the redistributed reward and the same optimal policies as $\tilde{\mathcal{P}}$:

**Theorem 1** (Arjona-Medina et al. (2019)). *Both the SDP $\tilde{\mathcal{P}}$ with delayed reward $\tilde{R}_{t+1}$ and the SDP $\mathcal{P}$ with redistributed reward $R_{t+1}$ have the same optimal policies.*

*Proof.* The proof can be found in (Arjona-Medina et al., 2019). □

The delay of rewards is captured by the *expected future rewards* $\kappa(m, t-1)$ at time $(t-1)$. $\kappa$ is defined as $\kappa(m, t-1) := \mathrm{E}_\pi[\sum_{\tau=0}^{m} R_{t+1+\tau} \mid s_{t-1}, a_{t-1}]$, that is, at time $(t-1)$ the expected sum of future rewards from $R_{t+1}$ to $R_{t+1+m}$ but not the immediate reward $R_t$. A reward redistribution is defined to be *optimal*, if $\kappa(T - t - 1, t) = 0$ for $0 \leqslant t \leqslant T - 1$, which is equivalent to $\mathrm{E}_\pi[R_{t+1} \mid s_{t-1}, a_{t-1}, s_t, a_t] = \tilde{q}^\pi(s_t, a_t) - \tilde{q}^\pi(s_{t-1}, a_{t-1})$:

**Theorem 2** (Arjona-Medina et al. (2019)). *We assume a delayed reward MDP $\tilde{\mathcal{P}}$, with episodic reward. A new SDP $\mathcal{P}$ is obtained by a second order Markov reward redistribution, which ensures that $\mathcal{P}$ is return-equivalent to $\tilde{\mathcal{P}}$. For a specific $\pi$, the following two statements are equivalent:*

*(I)* $\kappa(T - t - 1, t) = 0$, *i.e. the reward redistribution is optimal,*

*(II)* $\mathrm{E}_\pi\left[R_{t+1} \mid s_{t-1}, a_{t-1}, s_t, a_t\right] = \tilde{q}^\pi(s_t, a_t) - \tilde{q}^\pi(s_{t-1}, a_{t-1})$       (2)

*An optimal reward redistribution fulfills for $1 \leqslant t \leqslant T$ and $0 \leqslant m \leqslant T - t$: $\kappa(m, t - 1) = 0$.*

*Proof.* The proof can be found in (Arjona-Medina et al., 2019).      $\square$

This theorem shows that an optimal reward redistribution relies on steps $\tilde{q}^\pi(s_t, a_t) - \tilde{q}^\pi(s_{t-1}, a_{t-1})$ of the $Q$-function. Identifying the largest steps in the $Q$-function detects the largest rewards that have to be redistributed, which makes the largest progress towards obtaining an optimal reward redistribution.

**Corollary 1** (Higher order Markov reward redistribution optimality conditions)**.** *We assume a delayed reward MDP $\tilde{\mathcal{P}}$, with episodic reward. A new SDP $\mathcal{P}$ is obtained by a higher order Markov reward redistribution. The reward redistribution ensures that $\mathcal{P}$ is return-equivalent to $\tilde{\mathcal{P}}$. If for a specific $\pi$*

$$\mathrm{E}_\pi\left[R_{t+1} \mid s_{t-1}, a_{t-1}, s_t, a_t\right] = \tilde{q}^\pi(s_t, a_t) - \tilde{q}^\pi(s_{t-1}, a_{t-1}) \tag{3}$$

*holds, then the higher order reward redistribution $R_{t+1}$ is optimal, that is, $\kappa(T - t - 1, t) = 0$.*

*Proof.* The proof is just PART (II) of the proof of Theorem 2 in (Arjona-Medina et al., 2019). We repeat it here for completeness.

We assume that

$$\mathrm{E}_\pi\left[R_{t+1} \mid s_{t-1}, a_{t-1}, s_t, a_t\right] = h_t = \tilde{q}^\pi(s_t, a_t) - \tilde{q}^\pi(s_{t-1}, a_{t-1}), \tag{4}$$

where we abbreviate the expected $R_{t+1}$ by $h_t$:

$$\mathrm{E}_\pi\left[R_{t+1} \mid s_{t-1}, a_{t-1}, s_t, a_t\right] = h_t. \tag{5}$$

The expectations $\mathrm{E}_\pi\left[. \mid s_{t-1}, a_{t-1}\right]$ like $\mathrm{E}_\pi\left[\tilde{R}_{T+1} \mid s_{t-1}, a_{t-1}\right]$ are expectations over all episodes that contain the state-action pair $(s_{t-1}, a_{t-1})$ at time $t-1$. The expectations $\mathrm{E}_\pi\left[. \mid s_{t-1}, a_{t-1}, s_t, a_t\right]$ like $\mathrm{E}_\pi\left[\tilde{R}_{T+1} \mid s_{t-1}, a_{t-1}, s_t, a_t\right]$ are expectations over all episodes that contain the state-action pairs $(s_{t-1}, a_{t-1})$ at time $t - 1$ and $(s_t, a_t)$ at time $t$. The $Q$-values are defined as

$$\tilde{q}^\pi(s_t, a_t) = \mathrm{E}_\pi\left[\sum_{k=0}^{T-t} \tilde{R}_{t+k+1} \mid s_t, a_t\right] = \mathrm{E}_\pi\left[\tilde{R}_{T+1} \mid s_t, a_t\right], \tag{6}$$

$$q^\pi(s_t, a_t) = \mathrm{E}_\pi\left[\sum_{k=0}^{T-t} R_{t+k+1} \mid s_t, a_t\right], \tag{7}$$

which are expectations over all trajectories that contain $(s_t, a_t)$ at time $t$. Since $\tilde{\mathcal{P}}$ is Markov, for $\tilde{q}^\pi$ only the suffix trajectories beginning at $(s_t, a_t)$ enter the expectation.

The definition of $\kappa(m, t - 1)$ for $1 \leqslant t \leqslant T$ and $0 \leqslant m \leqslant T - t$ was $\kappa(m, t - 1) = \mathrm{E}_\pi\left[\sum_{\tau=0}^{m} R_{t+1+\tau} \mid s_{t-1}, a_{t-1}\right]$. **We have to proof $\kappa(T - t - 1, t) = 0$.**

First, we consider $m = 0$ and $1 \leqslant t \leqslant T$, therefore $\kappa(0, t - 1) = \mathrm{E}_\pi\left[R_{t+1} \mid s_{t-1}, a_{t-1}\right]$. Since the original MDP $\tilde{\mathcal{P}}$ has episodic reward, we have $\tilde{r}(s_{t-1}, a_{t-1}) = \mathrm{E}\left[\tilde{R}_t \mid s_{t-1}, a_{t-1}\right] = 0$ for $1 \leqslant t \leqslant T$. Therefore, we obtain:

$$\tilde{q}^\pi(s_{t-1}, a_{t-1}) = \tilde{r}(s_{t-1}, a_{t-1}) + \sum_{s_t, a_t} p(s_t, a_t \mid s_{t-1}, a_{t-1})\, \tilde{q}^\pi(s_t, a_t) \tag{8}$$

$$= \sum_{s_t, a_t} p(s_t, a_t \mid s_{t-1}, a_{t-1})\, \tilde{q}^\pi(s_t, a_t).$$

Using this equation we obtain for $1 \leqslant t \leqslant T$:

$$
\begin{aligned}
\kappa(0, t-1) &= \mathrm{E}_\pi \left[ R_{t+1} \mid s_{t-1}, a_{t-1} \right] & (9) \\
&= \mathrm{E}_{s_t, a_t} \left[ \tilde{q}^\pi(s_t, a_t) - \tilde{q}^\pi(s_{t-1}, a_{t-1}) \mid s_{t-1}, a_{t-1} \right] \\
&= \sum_{s_t, a_t} p(s_t, a_t \mid s_{t-1}, a_{t-1}) \left( \tilde{q}^\pi(s_t, a_t) - \tilde{q}^\pi(s_{t-1}, a_{t-1}) \right) \\
&= \tilde{q}^\pi(s_{t-1}, a_{t-1}) - \sum_{s_t, a_t} p(s_t, a_t \mid s_{t-1}, a_{t-1}) \, \tilde{q}^\pi(s_{t-1}, a_{t-1}) \\
&= \tilde{q}^\pi(s_{t-1}, a_{t-1}) - \tilde{q}^\pi(s_{t-1}, a_{t-1}) = 0 \,.
\end{aligned}
$$

Next, we consider the expectation of $\sum_{\tau=0}^m R_{t+1+\tau}$ for $1 \leqslant t \leqslant T$ and $1 \leqslant m \leqslant T - t$ (for $m > 0$)

$$
\begin{aligned}
\kappa(m, t-1) &= \mathrm{E}_\pi \left[ \sum_{\tau=0}^m R_{t+1+\tau} \mid s_{t-1}, a_{t-1} \right] & (10) \\
&= \mathrm{E}_\pi \left[ \sum_{\tau=0}^m \left( \tilde{q}^\pi(s_{\tau+t}, a_{\tau+t}) - \tilde{q}^\pi(s_{\tau+t-1}, a_{\tau+t-1}) \right) \mid s_{t-1}, a_{t-1} \right] \\
&= \mathrm{E}_\pi \left[ \tilde{q}^\pi(s_{t+m}, a_{t+m}) - \tilde{q}^\pi(s_{t-1}, a_{t-1}) \mid s_{t-1}, a_{t-1} \right] \\
&= \mathrm{E}_\pi \left[ \mathrm{E}_\pi \left[ \sum_{\tau=t+m}^T \tilde{R}_{\tau+1} \mid s_{t+m}, a_{t+m} \right] \mid s_{t-1}, a_{t-1} \right] \\
&\quad - \mathrm{E}_\pi \left[ \mathrm{E}_\pi \left[ \sum_{\tau=t-1}^T \tilde{R}_{\tau+1} \mid s_{t-1}, a_{t-1} \right] \mid s_{t-1}, a_{t-1} \right] \\
&= \mathrm{E}_\pi \left[ \tilde{R}_{T+1} \mid s_{t-1}, a_{t-1} \right] - \mathrm{E}_\pi \left[ \tilde{R}_{T+1} \mid s_{t-1}, a_{t-1} \right] \\
&= 0 \,.
\end{aligned}
$$

We used that $\tilde{R}_{t+1} = 0$ for $t < T$.

For the particualr cases $t = \tau + 1$ and $m = T - t = T - \tau - 1$ we have

$$
\kappa(T - \tau - 1, \tau) = 0 \,. \tag{11}
$$

That is exactly what we wanted to proof.

$\square$

Corollary 1 explicitly states that the optimality criterion ensures an optimal reward redistribution even if the reward redistribution is higher order Markov. For Align-RUDDER we may obtain a higher order Markov reward redistribution due to the profile alignment of the sub-sequences.

**Corollary 2** (Higher order Markov reward redistribution optimality representation). *We assume a delayed reward MDP $\tilde{\mathcal{P}}$, with episodic reward and that a new SDP $\mathcal{P}$ is obtained by a higher order Markov reward redistribution. The reward redistribution ensures that $\mathcal{P}$ is strictly return-equivalent to $\tilde{\mathcal{P}}$. We assume that the reward redistribuition is optimal, that is, $\kappa(T - t - 1, t) = 0$. If the condition*

$$
\mathrm{E}_\pi \left[ \sum_{\tau=0}^{T-t-1} R_{t+2+\tau} \mid s_t, a_t \right] = \mathrm{E}_\pi \left[ \sum_{\tau=0}^{T-t-1} R_{t+2+\tau} \mid s_0, a_0, \ldots, s_t, a_t \right] \tag{12}
$$

*holds, then*

$$
\mathrm{E}_\pi \left[ R_{t+1} \mid s_{t-1}, a_{t-1}, s_t, a_t \right] = \tilde{q}^\pi(s_t, a_t) - \tilde{q}^\pi(s_{t-1}, a_{t-1}) \,. \tag{13}
$$

*Proof.* By and large, the proof is PART (I) of the proof of Theorem 2 in (Arjona-Medina et al., 2019). We repeat it here for completeness.

We assume that the reward redistribution is optimal, that is,

$$\kappa(T - t - 1, t) = 0 . \tag{14}$$

We abbreviate the expected $R_{t+1}$ by $h_t$:

$$\mathrm{E}_\pi \left[ R_{t+1} \mid s_{t-1}, a_{t-1}, s_t, a_t \right] = h_t . \tag{15}$$

In (Arjona-Medina et al., 2019) Lemma A4 is as follows.

**Lemma 1.** *Two strictly return-equivalent SDPs $\tilde{\mathcal{P}}$ and $\mathcal{P}$ have the same expected return for each start state-action sub-sequence $(s_0, a_0, \ldots, s_t, a_t)$, $0 \leqslant t \leqslant T$:*

$$\mathrm{E}_\pi \left[ \tilde{G}_0 \mid s_0, a_0, \ldots, s_t, a_t \right] = \mathrm{E}_\pi \left[ G_0 \mid s_0, a_0, \ldots, s_t, a_t \right] . \tag{16}$$

The assumptions of Lemma 1 hold for for the delayed reward MDP $\tilde{\mathcal{P}}$ and the redistributed reward SDP $\mathcal{P}$, since a reward redistribution ensures strictly return-equivalent SDPs. Therefore for a given state-action sub-sequence $(s_0, a_0, \ldots, s_t, a_t)$, $0 \leqslant t \leqslant T$:

$$\mathrm{E}_\pi \left[ \tilde{G}_0 \mid s_0, a_0, \ldots, s_t, a_t \right] = \mathrm{E}_\pi \left[ G_0 \mid s_0, a_0, \ldots, s_t, a_t \right] \tag{17}$$

with $G_0 = \sum_{\tau=0}^{T} R_{\tau+1}$ and $\tilde{G}_0 = \tilde{R}_{T+1}$. The Markov property of the MDP $\tilde{\mathcal{P}}$ ensures that the future reward from $t + 1$ on is independent of the past sub-sequence $s_0, a_0, \ldots, s_{t-1}, a_{t-1}$:

$$\mathrm{E}_\pi \left[ \sum_{\tau=0}^{T-t} \tilde{R}_{t+1+\tau} \mid s_t, a_t \right] = \mathrm{E}_\pi \left[ \sum_{\tau=0}^{T-t} \tilde{R}_{t+1+\tau} \mid s_0, a_0, \ldots, s_t, a_t \right] . \tag{18}$$

According to Eq. (12), the future reward from $t + 2$ on is independent of the past sub-sequence $s_0, a_0, \ldots, s_{t-1}, a_{t-1}$:

$$\mathrm{E}_\pi \left[ \sum_{\tau=0}^{T-t-1} R_{t+2+\tau} \mid s_t, a_t \right] = \mathrm{E}_\pi \left[ \sum_{\tau=0}^{T-t-1} R_{t+2+\tau} \mid s_0, a_0, \ldots, s_t, a_t \right] . \tag{19}$$

Using these properties we obtain

$$
\begin{aligned}
\tilde{q}^{\pi}(s_t, a_t) &= \mathrm{E}_{\pi}\left[\sum_{\tau=0}^{T-t} \tilde{R}_{t+1+\tau} \mid s_t, a_t\right] \qquad (20) \\
&= \mathrm{E}_{\pi}\left[\sum_{\tau=0}^{T-t} \tilde{R}_{t+1+\tau} \mid s_0, a_0, \dots, s_t, a_t\right] \\
&= \mathrm{E}_{\pi}\left[\tilde{R}_{T+1} \mid s_0, a_0, \dots, s_t, a_t\right] \\
&= \mathrm{E}_{\pi}\left[\sum_{\tau=0}^{T} \tilde{R}_{\tau+1} \mid s_0, a_0, \dots, s_t, a_t\right] \\
&= \mathrm{E}_{\pi}\left[\tilde{G}_0 \mid s_0, a_0, \dots, s_t, a_t\right] \\
&= \mathrm{E}_{\pi}\left[G_0 \mid s_0, a_0, \dots, s_t, a_t\right] \\
&= \mathrm{E}_{\pi}\left[\sum_{\tau=0}^{T} R_{\tau+1} \mid s_0, a_0, \dots, s_t, a_t\right] \\
&= \mathrm{E}_{\pi}\left[\sum_{\tau=0}^{T-t-1} R_{t+2+\tau} \mid s_0, a_0, \dots, s_t, a_t\right] + \sum_{\tau=0}^{t} h_{\tau} \\
&= \mathrm{E}_{\pi}\left[\sum_{\tau=0}^{T-t-1} R_{t+2+\tau} \mid s_t, a_t\right] + \sum_{\tau=0}^{t} h_{\tau} \\
&= \kappa(T-t-1, t) + \sum_{\tau=0}^{t} h_{\tau} \\
&= \sum_{\tau=0}^{t} h_{\tau} \, .
\end{aligned}
$$

We used the optimality condition

$$
\kappa(T-t-1, t) = \mathrm{E}_{\pi}\left[\sum_{\tau=0}^{T-t-1} R_{t+2+\tau} \mid s_t, a_t\right] = 0 \, . \qquad (21)
$$

It follows that

$$
\mathrm{E}_{\pi}\left[R_{t+1} \mid s_{t-1}, a_{t-1}, s_t, a_t\right] = h_t = \tilde{q}^{\pi}(s_t, a_t) - \tilde{q}^{\pi}(s_{t-1}, a_{t-1}) \, . \qquad (22)
$$

This is exactly what we wanted to proof.

$\square$

This corollary shows that optimal reward redistributions can be expressed as difference of $Q$-values if Eq. (12) holds. Eq. (12) states that the past can be averaged out. However, there may exist optimal reward redistributions for which Eq. (12) does not hold.

If the reward redistribution is optimal, the $Q$-values of $\mathcal{P}$ are given by $q^{\pi}(s_t, a_t) = \tilde{q}^{\pi}(s_t, a_t) - \psi^{\pi}(s_t)$ and therefore $\tilde{\mathcal{P}}$ and $\mathcal{P}$ have the same advantage function:

**Theorem 3** (Arjona-Medina et al. (2019))**.** *If the reward redistribution is optimal, then the $Q$-values of the SDP $\mathcal{P}$ are $q^{\pi}(s_t, a_t) = r(s_t, a_t)$ and*

$$
q^{\pi}(s_t, a_t) = \tilde{q}^{\pi}(s_t, a_t) - \mathrm{E}_{s_{t-1}, a_{t-1}}\left[\tilde{q}^{\pi}(s_{t-1}, a_{t-1}) \mid s_t\right] = \tilde{q}^{\pi}(s_t, a_t) - \psi^{\pi}(s_t) \, . \qquad (23)
$$

*The SDP $\mathcal{P}$ and the original MDP $\tilde{\mathcal{P}}$ have the same advantage function.*

*Proof.* The proof can be found in (Arjona-Medina et al., 2019). $\square$

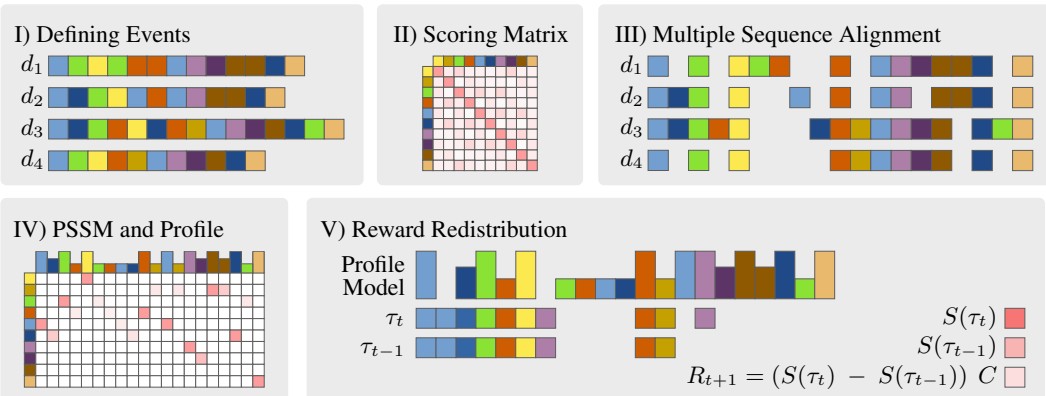

Figure A.1: The five steps of Align-RUDDER's reward redistribution. **(I)** shows several demonstrations, where each demonstration is composed of a sequence of events. Events are defined as difference of state-actions or clusters thereof. **Step (II)** depicts a scoring matrix, which we construct using event probabilities from demonstrations. With demonstrations and the scoring matrix we then perform MSA in **step (III)**. In **step (IV)** to construct the profile model and PSSM from the alignment. These are then used to align a new sequence to the model in **step (V)**, one timestep at a time. The differences in alignment score are then used to redistribute reward for this new sequence.

For an optimal reward redistribution only the expectation of the immediate reward $r(s_t, a_t) = \mathrm{E}_\pi \left[ R_{t+1} \mid s_t, a_t \right]$ must be estimated. This considerably simplifies learning.

**Learning methods according to Arjona-Medina et al. (2019).** The redistributed reward serves as reward for a subsequent learning method, which can be Type A, B, and C as described in Arjona-Medina et al. (2019). Type A methods estimate the $Q$-values. They can be estimated directly according to Eq. (23) assuming an optimal redistribution (Type A variant i). $Q$-values can be corrected for a non-optimal reward redistribution by additionally estimating $\kappa$ (Type A variant ii). $Q$-value estimation can use eligibility traces (Type A variant iii). Type B methods use the redistributed rewards for policy gradients like Proximal Policy Optimization (PPO) Schulman et al. (2018). Type C methods use TD learning like $Q$-learning Watkins (1989), where immediate and future reward must be drawn together as typically done. For all these learning methods, demonstrations can be used for initialization (e.g. experience replay buffer) or pre-training (e.g. policy network with behavioral cloning). Recently, the convergence of RUDDER learning methods has been proven under commonly used assumptions (Holzleitner et al., 2020).

**Non-optimal reward redistribution and Align-RUDDER.** According to Theorem 1, non-optimal reward redistributions do not change the optimal policies. The value $\kappa(T - t - 1, t)$ measures the remaining delayed reward. The smaller $\kappa$ is, the faster is the learning process. For Monte Carlo (MC) estimates, smaller $\kappa$ reduces the variance of the future rewards, and, therefore the variance of the estimation. For temporal difference (TD) estimates, smaller $\kappa$ reduces the amount of information that has to flow back. Align-RUDDER dramatically reduces the amount of delayed rewards by identifying key events via multiple sequence alignment, to which reward is redistributed. For an episodic MDP, a reward that is redistributed to time $t$ reduces all $\kappa(m, \tau)$ with $t \leqslant \tau < T$ by the expectation of the reward. Therefore, in most cases Align-RUDDER makes $\kappa$-values much smaller.

### A.3 THE FIVE STEPS OF ALIGN-RUDDER'S REWARD REDISTRIBUTION

The new reward redistribution approach consists of five steps, see Fig. A.1: (I) Define events to turn episodes of state-action sequences into sequences of events. (II) Determine an alignment scoring scheme, so that relevant events are aligned to each other. (III) Perform a multiple sequence alignment (MSA) of the demonstrations. (IV) Compute the profile model and the PSSM. (V) Redistribute the reward: Each sub-sequence $\tau_t$ of a new episode $\tau$ is aligned to the profile. The redistributed reward $R_{t+1}$ is proportional to the difference of scores $S$ based on the PSSM given in step (IV), i.e. $R_{t+1} \propto S(\tau_t) - S(\tau_{t-1})$.

**(I) Defining Events.** Alignment techniques assume that sequences consist of few symbols, e.g. about 20 symbols, the events. It is crucial to keep the number of events small in order to increase the difference between a random alignment and an alignment of demonstrations. If there are many events, then two demonstrations might have few events that can be matched, which cannot be well distinguished from random alignments. This effect is known in bioinformatics as "Inconsistency of Maximum Parsimony" (Felsenstein, 1978). The events can be the original state-action pairs, clusters thereof, or other representations of state-action pairs, e.g. indicating changes of inventory, health, energy, skills etc. In general, we define events as a cluster of states or state-actions. A sequence of events is obtained from a state-action sequence by substituting states or state-actions by their cluster identifier. In order to cluster states, a similarity measure between them is required. We suggest to use the "successor representation" (Dayan, 1993) of the states, which gives a similarity matrix based on how connected two states are given a policy. Successor representation have been used before (Machado et al., 2017; Ramesh et al., 2019) to obtain important events, for option learning. For computing the successor representation, we use the demonstrations combined with state-action sequences generated by a random policy. For high dimensional state spaces "successor features" (Barreto et al., 2017) can be used. We use similarity-based clustering methods like affinity propagation (AP) (Frey & Dueck, 2007). For AP the similarity matrix does not have to be symmetric and the number of clusters need not be known. State action pairs $(s, a)$ are mapped to events $e$.

**(II) Determining the Alignment Scoring System.** Alignment algorithms distinguish similar sequences from dissimilar sequences using a scoring system. A scoring matrix $\mathbb{S}$ has entries $\mathbb{s}_{i,j}$ that give the score for aligning event $i$ with $j$. The MSA score $S_{\text{MSA}}$ of a multiple sequence alignment is the sum of all pairwise scores: $S_{\text{MSA}} = \sum_{i,j,i<j} \sum_{t=0}^{L} \mathbb{s}_{x_{i,t}, x_{j,t}}$, where $x_{i,t}$ means that event $x_{i,t}$ is at position $t$ for sequence $\tau_i = e_{i,0:T}$ in the alignment, analog for $x_{j,t}$ and the sequence $\tau_j = e_{j,0:T}$, and $L$ is the alignment length. Note that $L \geq T$ and $x_{i,t} \neq e_{i,t}$, since gaps are present in the alignment. In the alignment, events should have the same probability of being aligned as they would have if we know the strategy and align demonstrations accordingly. The theory of high scoring segments gives a scoring scheme with these alignment probabilities (Karlin & Altschul, 1990; Karlin et al., 1990; Altschul et al., 1990). Event $i$ is observed with probability $p_i$ in the demonstrations, therefore a random alignment aligns event $i$ with $j$ with probability $p_i p_j$. An alignment algorithm maximizes the MSA score $S_{\text{MSA}}$ and, thereby, aligns events $i$ and $j$ with probability $q_{ij}$ for demonstrations. High values of $q_{ij}$ means that the MSA often aligns events $i$ and $j$ in the demonstrations using the scoring matrix $\mathbb{S}$ with entries $\mathbb{s}_{i,j}$. According to Theorem 2 and Equation [3] in Karlin & Altschul (1990), asymptotically with the sequence length, we have $\mathbb{s}_{i,j} = \ln(q_{ij}/(p_i p_j))/\lambda^*$, where $\lambda^*$ is the unique positive root of $\sum_{i=1,j=1}^{n,n} p_i p_j \exp(\lambda \mathbb{s}_{i,j}) = 1$ (Equation [4] in Karlin & Altschul (1990)).

We can now choose a desired probability $q_{ij}$ and then compute the scoring matrix $\mathbb{S}$ with entries $\mathbb{s}_{i,j}$. High values of $q_{ij}$ should indicate relevant events for the strategy. A priori, we only know that a relevant event should be aligned to itself, while we do not know which events are relevant. Therefore we set $q_{ij}$ to large values for every $i = j$ and to low values for $i \neq j$. Concretely, we set $q_{ij} = p_i - \epsilon$ for $i = j$ and $q_{ij} = \epsilon/(n-1)$ for $i \neq j$, where $n$ is the number of different possible events. Events with smaller $p_i$ receive a higher score $\mathbb{s}_{i,i}$ when aligned to themselves since this self-match is less often observed when randomly matching events ($p_i p_i$ is the probability of a random self-match). Any prior knowledge about events should be incorporated into $q_{ij}$.

**(III) Multiple sequence alignment (MSA).** MSA first produces pairwise alignments between all demonstrations. Then, a guiding tree (agglomerative hierarchical clustering) is produced via hierarchical clustering sequences, according to their pairwise alignment scores. Demonstrations which follow the same strategy appear in the same cluster in the guiding tree. Each cluster is aligned separately via MSA to address different strategies. However, if there is not a cluster of demonstrations, then the alignment will fail. MSA methods like ClustalW (Thompson et al., 1994) or MUSCLE (Edgar, 2004) can be used.

**(IV) Position-Specific Scoring Matrix (PSSM) and Profile.** From the final alignment, we construct a) an MSA profile (column-wise event frequencies $q_{i,j}$) and b) a PSSM (Stormo et al., 1982) which is used for aligning new sequences to the profile of the MSA. To compute the PSSM (column-wise scores $\mathbb{s}_{i,t}$), we apply Theorem 2 and Equation [3] in Karlin & Altschul (1990). Event $i$ is observed with probability $p_i$ in the data. For each position $t$ in the alignment, we compute $q_{i,t}$, which indicates the frequency of event $i$ at position $t$. The PSSM is $\mathbb{s}_{i,t} = \ln(q_{i,t}/p_i)/\lambda_t^*$, where $\lambda_t^*$ is the single unique positive root of $\sum_{i=1}^{n} p_i \exp(\lambda \mathbb{s}_{i,t}) = 1$ (Equation [1] in Karlin & Altschul (1990)). If we

align a new sequence that follows the underlying strategy (a new demonstration) to the profile model, we would see that event $i$ is aligned to position $t$ in the profile with probability $q_{i,t}$.

**(V) Reward Redistribution.** The reward redistribution is based on the profile model. A sequence $\tau = e_{0:T}$ ($e_t$ is event at position $t$) is aligned to the profile, which gives the score $S(\tau) = \sum_{t=0}^{L} \mathbb{s}_{x_t,t}$. Here, $\mathbb{s}_{i,t}$ is the alignment score for event $i$ and $x_t$ is the event of $\tau$ at position $t$ in the alignment. $L$ is the profile length, where $L \geq T$ and $x_t \neq e_t$, because of gaps in the alignment. If $\tau_t = e_{0:t}$ is the prefix sequence of $\tau$ of length $t + 1$, then the reward redistribution $R_{t+1}$ for $0 \leqslant t \leqslant T$ is

$$R_{t+1} = (S(\tau_t) - S(\tau_{t-1}))\,C = g((s,a)_{0:t}) - g((s,a)_{0:t-1}), \; R_{T+2} = \tilde{G}_0 - \sum_{t=0}^{T} R_{t+1}, \tag{24}$$

where $C = \mathrm{E}_{\mathrm{demo}}\left[\tilde{G}_0\right] \big/ \mathrm{E}_{\mathrm{demo}}\left[\sum_{t=0}^{T} S(\tau_t) - S(\tau_{t-1})\right]$ and $\tilde{G}_0 = \sum_{t=0}^{T} \tilde{R}_{t+1}$ is the original return of the sequence $\tau$ and $S(\tau_{-1}) = 0$. $\mathrm{E}_{\mathrm{demo}}$ is the expectation over demonstrations, and $C$ scales $R_{t+1}$ to the range of $\tilde{G}_0$. $R_{T+2}$ is the correction of the redistributed reward (Arjona-Medina et al., 2019), with zero expectation for demonstrations: $\mathrm{E}_{\mathrm{demo}}[R_{T+2}] = 0$. Since $\tau_t = e_{0:t}$ and $e_t = f(s_t, a_t)$, we can set $g((s,a)_{0:t}) = S(\tau_t)C$. We ensure strict return equivalence, since $G_0 = \sum_{t=0}^{T+1} R_{t+1} = \tilde{G}_0$. The redistributed reward depends only on the past, that is, $R_{t+1} = h((s,a)_{0:t})$. For computational efficiency, the alignment of $\tau_{t-1}$ can be extended to one for $\tau_t$, like exact matches are extended to high-scoring sequence pairs with the BLAST algorithm (Altschul et al., 1990; 1997).

*Sub-tasks.* The reward redistribution identifies sub-tasks, which are alignment positions with high redistributed reward. It also determines the terminal states and automatically assigns reward for solving the sub-tasks. However, reward redistribution and Align-RUDDER cannot guarantee that the reward is Markov. For redistributed reward that is Markov, the option framework (Sutton et al., 1999), the MAXQ framework (Dietterich, 2000), or recursive composition of option models (Silver & Ciosek, 2012) can be used as subsequent approaches to hierarchical reinforcement learning.

## A.4 SEQUENCE ALIGNMENT

In bioinformatics, sequence alignment identifies regions of significant similarity among different biological sequences to establish evolutionary relationships between those sequences. In 1970, Needleman and Wunsch proposed a global alignment method based on dynamic programming (Needleman & Wunsch, 1970). This approach ensures the best possible alignment given a substitution matrix, such as PAM (Dayhoff, 1978) or BLOSUM(Henikoff & Henikoff, 1992), and other parameters to penalize gaps in the alignment. The method of Needlemann and Wunsch is of $O(mn)$ complexity both in memory and time, which could be prohibitive in long sequences like genomes. An optimization of this method by Hirschberg (1975), reduces memory to $O(m + n)$, but still requires $O(mn)$ time.

Later, Smith and Waterman developed a local alignment method for sequences (Smith & Waterman, 1981). It is a variation of Needleman and Wunsch's method, keeping the substitution matrix and the gap-scoring scheme but setting cells in the similarity matrix with negative scores to zero. The complexity for this algorithm is of $O(n^2 M)$. Osamu Gotoh published an optimization of this method, running in $O(mn)$ runtime (Gotoh, 1982).

The main difference between both methods is the following:

- The global alignment method by Needleman and Wunsch aligns the sequences fixing the first and the last position of both sequences. It attempts to align every symbol in the sequence, allowing some gaps, but the main purpose is to get a global alignment. This is especially useful when the two sequences are highly similar. For instance:

```
ATCGGATCGACTGGCTAGATCATCGCTGG
CGAGCATC-ACTGTCT-GATCGACCTTAG
*  ***  ****  ** ****    *    *  *
```

- As an alternative to global methods, the local method of Smith and Waterman aligns the sequences with a higher degree of freedom, allowing the alignment to start or end with gaps.

This is extremely useful when the two sequences are substantially dissimilar in general but suspected of having a highly related sub region.

```
ATCAAGGAGATCATCGCTGGACTGAGTGGCT----ACGTGGTATGT
ATC----CGATCATCGCTGG-CTGATCGACCTTCTACGT-------
***       *********** ****   *  *      ****
```

**A.4.0.1 Multiple Sequence Alignment algorithms.** The sequence alignment algorithms by Needleman and Wunsch and Smith and Waterman are limited to aligning two sequences. The approaches for generalizing these algorithms to multiple sequences can be classified into four categories:

- Exact methods (Wang & Jiang, 1994).
- Progressive methods: ClustalW (Thompson et al., 1994), Clustal Omega (Sievers et al., 2014), T-Coffee (Notredame et al., 2000).
- Iterative and search algorithms: DIALIGN (Morgenstern, 2004), MultiAlign (Corpet, 1988).
- Local methods: eMOTIF (Mccammon & Wolynes, 1998), PROSITE (Bairoch & Bucher, 1994).

For more details, visit *Sequence Comparison: Theory and methods* (Chao & Zhang, 2009).

In our experiments, we use ClustalW from Biopython (Cock et al., 2009) with the following parameters:

```
clustalw2 -ALIGN -CLUSTERING=UPGMA -NEGATIVE " \
        "-INFILE={infile} -OUTFILE={outfile} " \
        "-PWMATRIX={scores} -PWGAPOPEN=0 -PWGAPEXT=0 " \
        "-MATRIX={scores} -GAPOPEN=0 -GAPEXT=0 -CASE=UPPER " \
        "-NOPGAP -NOHGAP -MAXDIV=0 -ENDGAPS -NOVGAP " \
        "-NEWTREE={outputtree} -TYPE=PROTEIN -OUTPUT=GDE
```

where the PWMATRIX and MATRIX are computed according to step (II) in Sec. 3 of the main paper.

## A.5 EXTENDED RELATED WORK

Align-RUDDER allows to identify sub-goals and sub-tasks, therefore it is related to hierarchical reinforcement learning (HRL) approaches like the option framework (Sutton et al., 1999), the MAXQ framework (Dietterich, 2000), or the recursive composition of option models (Silver & Ciosek, 2012). However, these methods do not address the problem of finding good options, good sub-goals, or good sub-tasks. Methods to learn good options have been proposed. Frequently observed states in solutions are chosen as targets (Stolle & Precup, 2002). Gradient-based approaches improving the termination function for options (Comanici & Precup, 2010; Mankowitz et al., 2016). Policy gradient optimized a unified policy consisting of intra-option policies, option termination conditions, and an option selection policy (Levy & Shimkin, 2012). Parametrized options are learned by treating the termination functions as hidden variables and using expectation maximization (Daniel et al., 2016). Intrinsic rewards are used to learn the policies within options, and extrinsic rewards to learn the policy over options (Kulkarni et al., 2016). Options have been jointly learned with an associated policy using the policy gradient theorem for options (Bacon et al., 2017). A slow time-scale manager module learns sub-goals that are achieved by fast time-scale worker (Vezhnevets et al., 2017).

Next, we relate Align-RUDDER to imitation learning and trajectory matching. Imitation learning aims at learning a behavior close to the data generating policy by matching the trajectories of single demonstrations. In contrast, Align-RUDDER does not try to match single trajectories but identifies relevant events that are shared among successful demonstrations. In complex tasks like MineCraft trajectory matching fails, since large state spaces do not allow to match one of the few demonstrations. However, relevant events can still be matched as they appear in most demonstrations, therefore Align-RUDDER excels in such complex tasks.

## A.6 ARTIFICIAL TASK EXPERIMENTS

This section provides additional information that supports the results reported in the main paper for Artificial Tasks (I) and (II).

### A.6.1 HYPERPARAMETER SELECTION

For (BC)+$Q$-Learning and Align-RUDDER, we performed a grid search to select the learning rate from the following values $[0.1, 0.05, 0.01]$. We used 20 different seeds for each value and each number of demonstrations and then selected the setting with the highest success for all number of demonstrations. The final learning rate for (BC)+$Q$-Learning and DQfD is $0.01$ and for Align-RUDDER it is $0.1$.

For DQfD, we set the experience buffer size to $30,000$ and the number of experiences sampled at every timestep to $10$. The DQfD loss weights are set to $0.01$, $0.01$ and $1.0$ for the $Q$-learning loss term, $n$-step loss term and the expert loss respectively during pre-training. During online learning, we change the loss terms to $1.0$, $1.0$ and $0.01$ for the $Q$-learning loss term, $n$-step loss term and the expert loss term. This was necessary to enable faster learning for DQfD. The expert action margin is $0.8$.

For successor representation, we use a learning rate of $0.1$ and a gamma of $0.99$. We update the successor table multiple times using the same transition (state, action, next state) from the demonstration.

For affinity propagation, we use a damping factor of $0.5$ and set the maximum number of iterations to $1000$. Furthermore, if we obtain more than 15 clusters, then we combine clusters based on the similarity of the cluster centers.

### A.6.2 FIGURES

Figure A.5 shows sample trajectories in the FourRooms and EightRooms environment, with the initial and target positions marked in red and green respectively. Figure A.2 shows the clusters after performing clustering with Affinity Propagation using the successor representation with 25 demonstrations and an environment with 1% stochasticity on the transitions. Different colors indicate different clusters. Figures A.3 and A.4 show clusters for different environment settings. Figure A.3 shows clusters when using 10 demonstrations and for Figure A.4 environments with 5% stochasticity on transitions was used. Figure A.6 shows the reward redistribution for the given example trajectories in the FourRooms and EightRooms environments.

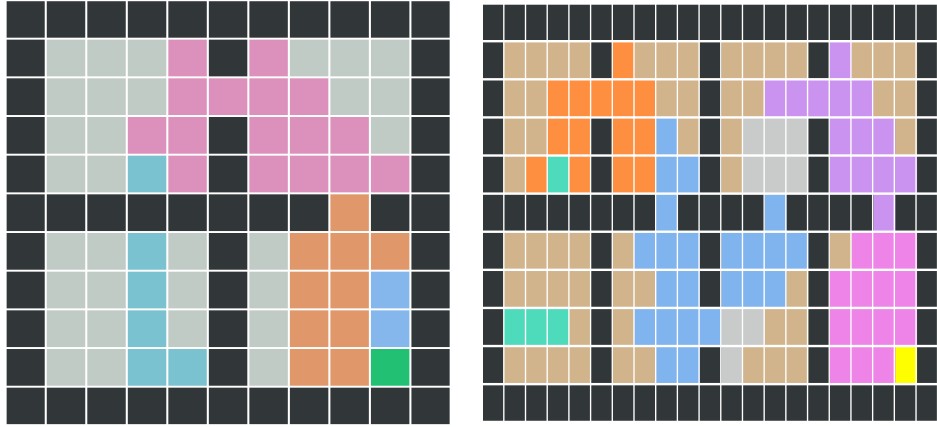

Figure A.2: Examples of clusters formed in the FourRooms (left) and EightRooms (right) environment with 1% stochasticity on the transitions after performing clustering with Affinity Propagation using the successor representation with 25 demonstrations. Different colors represent different clusters.

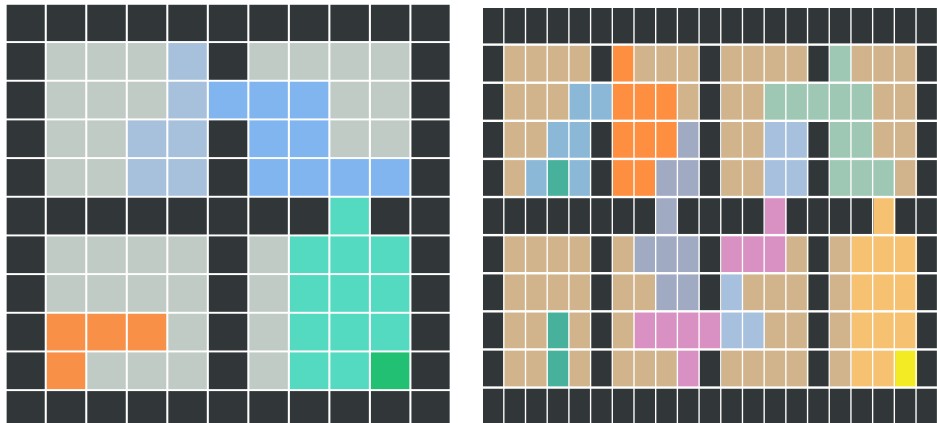

Figure A.3: Examples of clusters formed in the FourRooms (left) and EightRooms (right) environment with 1% stochasticity on the transitions after performing clustering with Affinity Propagation using the successor representation with 10 demonstrations. Different colors represent different clusters.

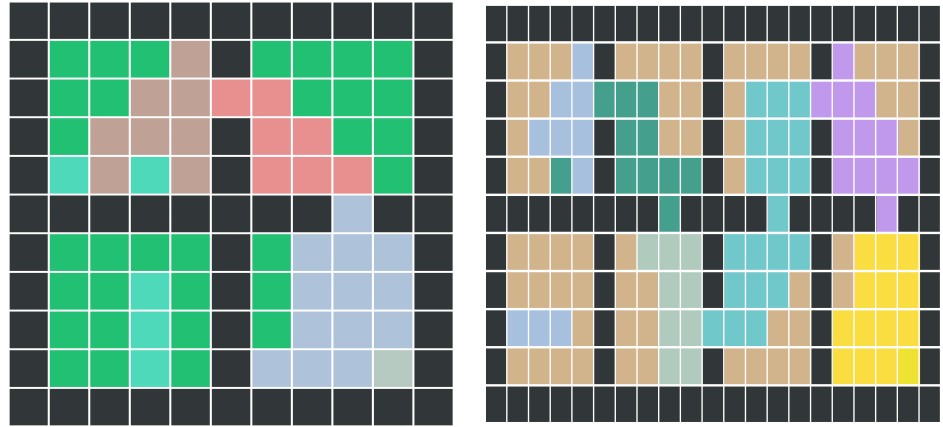

Figure A.4: Examples of clusters formed in the FourRooms (left) and EightRooms (right) environment with 5% stochasticity on the transitions after performing clustering with Affinity Propagation using the successor representation with 25 demonstrations. Different colors represent different clusters.

### A.6.3 ARTIFICIAL TASK P-VALUES

Tables A.1 and A.2 show the $p$-values obtained by performing a Mann-Whitney-U test between Align-RUDDER and BC+$Q$-Learning and DQfD respectively.

|  | 2 | 5 | 10 | 50 | 100 |
|---|---|---|---|---|---|
| Align-RUDDER vs. BC+$Q$-Learn. | 8.8e-31 | 2.8e-30 | 1.1e-09 | 3.5e-01 | 1.6e-01 |
| Align-RUDDER vs. SQIL | 3.6e-39 | 5.2e-39 | 3.1e-37 | 8.6e-36 | 1.9e-36 |
| Align-RUDDER vs. DQfD | 2.7e-29 | 4.3e-30 | 1.3e-32 | 1.0e+00 | 1.0e+00 |
| Align-RUDDER vs. RUDDER (LSTM) | 1.9e-31 | 1.9e-27 | 3.7e-20 | 1.7e-15 | 7.3e-01 |

Table A.1: $p$-values for Artificial Task (I), FourRooms, obtained by performing a Mann-Whitney-U test.

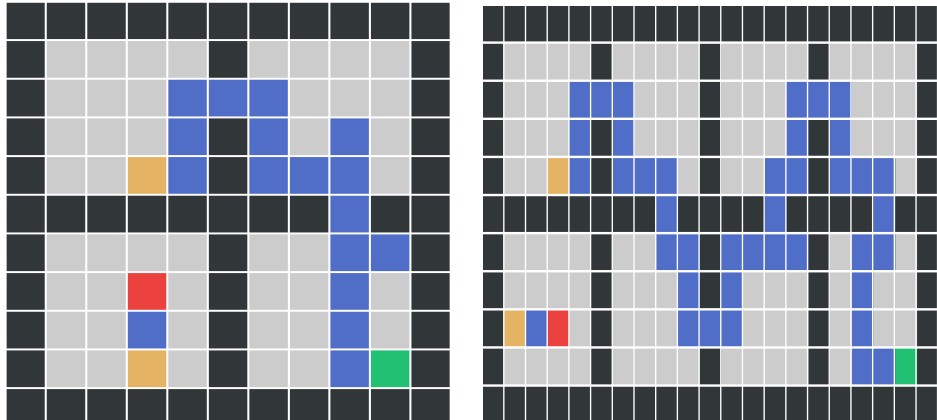

Figure A.5: Exemplary trajectories in the FourRooms (left) and EightRooms (right) environments. Initial position is indicated in red, the portal between the first and second room in yellow and the goal in green.

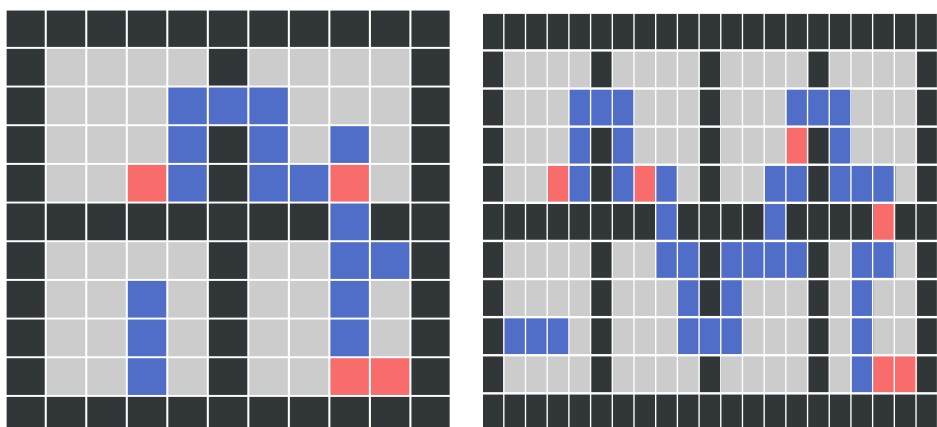

Figure A.6: Reward redistribution for the above trajectories in the FourRooms (left) and EightRooms (right) environments.

### A.6.4 STOCHASTIC ENVIRONMENTS

Figure A.7 shows results for the FourRooms environment with different levels of stochasticity (5%, 10%, 15%, 25% and 40%) on the transitions. Figure A.8 shows results for the EightRooms environment with different levels of stochasticity (5% and 10%) on the transitions.

|  | 2 | 5 | 10 | 50 | 100 |
|---|---|---|---|---|---|
| Align-RUDDER vs. BC+$Q$-Learn. | 4.5e-20 | 1.3e-34 | 4.9e-25 | 3.7e-01 | 6.1e-01 |
| Align-RUDDER vs. SQIL | 1.8e-37 | 2.8e-39 | 1.9e-36 | 1.7e-35 | 1.9e-37 |
| Align-RUDDER vs. DQfD | 1.2e-08 | 8.9e-20 | 5.6e-31 | 1.0e+00 | 1.0e+00 |
| Align-RUDDER vs. RUDDER (LSTM) | 1.2e-29 | 1.3e-34 | 3.9e-31 | 8.7e-22 | 1.0e-18 |

Table A.2: $p$-values for Artificial Task (II), EightRooms, obtained by performing a Mann-Whitney-U test.

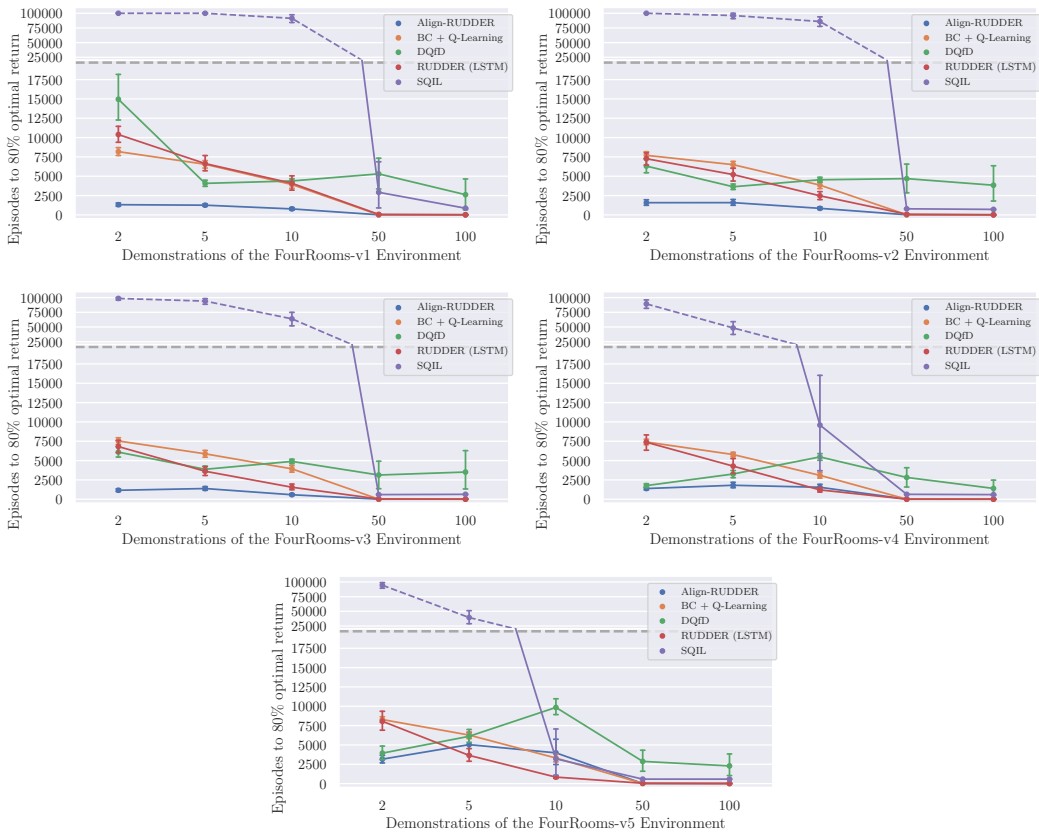

Figure A.7: Comparison of Align-RUDDER and other methods on Task (I) (FourRooms) with increasing levels of stochasticity (from top left to bottom: 5%, 10%, 15%, 25% and 40%). Results are the average over 50 trials.

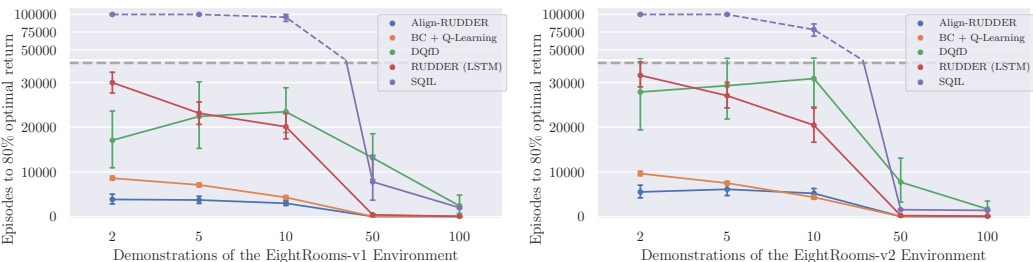

Figure A.8: Comparison of Align-RUDDER and other methods on Task (II) (EightRooms) with increasing levels of stochasticity (from top left to bottom: 5%, 10%). Results are the average over 50 trials.

### A.6.5 CHANGING NUMBER OF CLUSTERS

We use Affinity Propagation for clustering, and do not set the number of clusters. Although, we set the max number of clusters allowed. If Affinity propagation results in more clusters, they are combined and reduced to the maximum clusters allowed. This is necessary due to the limitations of the underlying alignment library we are using. For the experiments on FourRooms and EightRooms in the main paper, we fix the max number of clusters to 15.

We conduct an experimental study on how changing the max number of clusters changes the performance of Align-RUDDER on the FourRooms environment. The results are in table A.3.

| Max. # of Clusters | 2 | 5 | 10 | 50 | 100 |
|---|---|---|---|---|---|
| 2 | 5782.1 | 2378 | 7624 | 18 | 14 |
| 5 | 4462.1 | 1277 | 7892 | 19 | 14 |
| 8 | 985 | 1417 | 1372 | 19 | 14 |
| 10 | 985 | 1417 | 1372 | 19 | 14 |
| 12 | 985 | 1417 | 1372 | 19 | 14 |
| 15 | 985 | 1417 | 1372 | 19 | 14 |

Table A.3: Results for different numbers of clusters for the FourRooms artificial task are shown in the table. It shows the number episodes required to reach 80% optimal return, using a demonstrations given in column headers. These results are averaged over 10 seeds. For the results we report in the paper we set the maximum number of clusters to 15, the results show that even when reducing the number of clusters to 8, results stay similar. We only see worse performance for when only allowing 2 or 5 clusters.

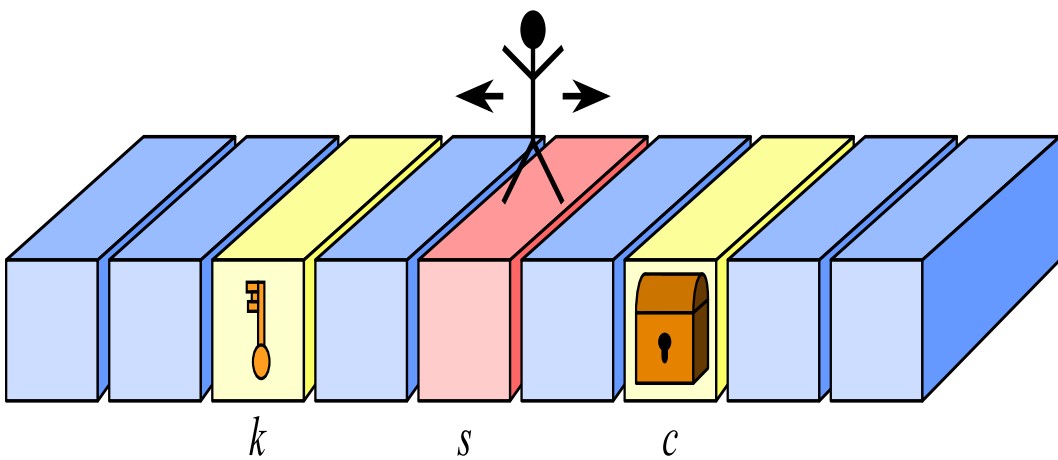

Figure A.9: The agent has to collect the key and then open the chest, to get a positive reward at the last timestep. The environment episode runs for a fixed 32 timesteps.

### A.6.6 KEY-EVENT DETECTION

**1D key-chest environment.** We use a *1D key-chest environment* to show the effectiveness of sequence alignment in a low data regime compared to an LSTM model. The agent has to collect the key and then open the chest, to get a positive reward at the last timestep. See Appendix Fig. A.9 for a schematic representation of the environment. As the key-events (important state-action pairs) in this environment are known we can compute the *key-event detection rate* of a reward redistribution model. A key event is detected if the redistributed reward of an important state-action pair is larger than the average redistributed reward in the sequence. We train the reward redistribution models with 2, 5 and 10 training episodes and test on 1000 test episodes, averaged over 10 trials. Align-RUDDER significantly outperforms LSTM (RUDDER) for detecting these key events in all cases, with an average key-event detection rate of 0.96 for sequence alignment vs. 0.46 for the LSTM models over all dataset sizes. See Appendix Fig. A.10 for the detailed results.

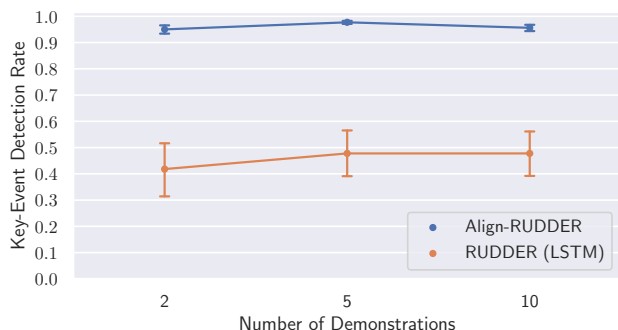

Figure A.10: We use Align-RUDDER and RUDDER to detect key events for Key-Chest environment, where we already know which state-action pairs are important for return. A key event is detected if the redistributed reward at an important state-action is larger than the average redistributed reward in the sequence. We test on 1000 test episodes and average over 10 trials. Align-RUDDER outperforms LSTM (RUDDER) for detecting these key events.

## A.7 MINECRAFT EXPERIMENTS

In this section we explain in detail the implementation of Align-RUDDER for solving the task *ObtainDiamond*.

### A.7.1 MINECRAFT

We show that our approach can be applied to complex tasks by evaluating it on the MineRL Minecraft dataset (Guss et al., 2019b). This dataset provides a large collection of demonstrations from human players solving six different tasks in the sandbox game MineCraft. In addition to the human demonstrations the MineRL dataset also provides an OpenAI-Gym wrapper for MineCraft. The dataset includes demonstrations for the following tasks:

- navigating to target location following a compass,
- collecting wood by chopping trees,
- obtaining an item by collecting resources and crafting, and
- free play "survival" where the player is free to choose his own goal.

The demonstrations include the video showing the players' view (without user interface), the players' inventory at every time step and the actions performed by the player. We focus on the third task of obtaining a target item, namely a diamond. This task is very challenging as it is necessary to obtain several different resources and tools and has been the focus of a challenge (Guss et al., 2019a) at NeurIPS'19. By the end of this challenge no entry was able to obtain the diamond.

We show that our method is well suited for solving the task of obtaining the diamond, which can be decomposed into sub-tasks by reward redistribution after aligning successful demonstrations.

### A.7.2 RELATED WORK AND STEPS TOWARDS A GENERAL AGENT

In the following, we review two approaches Skrynnik et al. (2019); Scheller et al. (2020) where more details are available and compare them with our approach.

Skrynnik et al. (2019) address the problem with a TD based hierarchical Deep $Q$-Network (DQN) and by utilizing the hierarchical structure of expert trajectories by extracting sequences of meta-actions and sub-goals. This approach allowed them to achieve the *1st* place in the official NeurIPS'19 MineRL challenge (Skrynnik et al., 2019). In terms of pre-processing, our approaches have in common that both rely on frame skipping and action space discretization. However, they reduce the action space to ten distinct joint environment actions (e.g. *move camera & attack*) and treat inventory actions separately by executing a sequence of semantic actions. We aim at taking a next step towards a more general agent by introducing an action space preserving the agent's full freedom of action in

the environment (more details are provided below). This allows us to avoid the distinction between item (environment) and semantic (inventory) agents and to train identically structured agents in the same fashion regardless of facing a mining, crafting, placing or smelting sub-task. Skrynnik et al. (2019) extract a sub-task chain by separately examining each expert trajectory and by considering the time of appearance of items in the inventory in chronological order. For agent training their approach follows a heuristic where they distinguish between collecting the item *log* and all remaining items. The *log*-agent is trained by starting with the *TreeChop* expert trajectories and then gradually injecting trajectories collected from interactions with the environment into the DQN's replay buffer. For the remaining items they rely on the expert data of *ObtainIronPickaxeDense* and imitation learning. Given our proposed sequence alignment and reward redistribution methodology we are able to avoid this shift in training paradigm and to leverage all available training data (*ObtainDiamond*, *ObtainIronPickaxe* and *TreeChop*) at the same time. In short, we collect all expert trajectories in one pool, perform sequence alignment yielding a common diamond consensus along with the corresponding reward redistribution and the respective sub-task sequences. Given this restructuring of the problem into local sub-problems with redistributed reward all sub-task agents are then trained in the same fashion (e.g. imitation learning followed by RL-based fine-tuning). Reward redistribution guarantees that the optimal policies are preserved (Arjona-Medina et al., 2019).

Scheller et al. (2020) achieved the *3rd* place on the official leader board following a different line of research and addressed the problem with a single *end-to-end* off-policy IMPALA (Espeholt et al., 2018) actor-critic agent, again utilizing experience replay to incorporate the expert data (Scheller et al., 2020). To prevent catastrophic forgetting of the behavior for later, less frequent sub-tasks they introduce value clipping and apply CLEAR (Rolnick et al., 2019) to both, policy and value networks. Treating the entire problem as a whole is already the main distinguishable feature compared to our method. To deal with long trajectories they rely on a trainable special form of frame skipping where the agent also has to predict how many frames to skip in each situation. This helps to reduce the effective length (step count) of the respective expert trajectories. In contrast to the approach of (Scheller et al., 2020) we rely on a constant frame skip irrespective of the states and actions we are facing. Finally, there are also several common features including:

1. a supervised BC pre-training stage prior to RL fine-tuning,
2. separate networks for policy and value function,
3. independent action heads on top of a sub-sequence LSTM,
4. presenting the inventory state in a certain form to the agent and
5. applying a value-function-burn-in prior to RL fine-tuning.

### A.7.3   THE FIVE STEPS OF ALIGN-RUDDER DEMONSTRATED ON MINECRAFT

In this subsection, we give an example of the five steps of Align-RUDDER using demonstrations from the MineRL *ObtainDiamond* task. Figures A.11 to A.15 illustrate these steps.

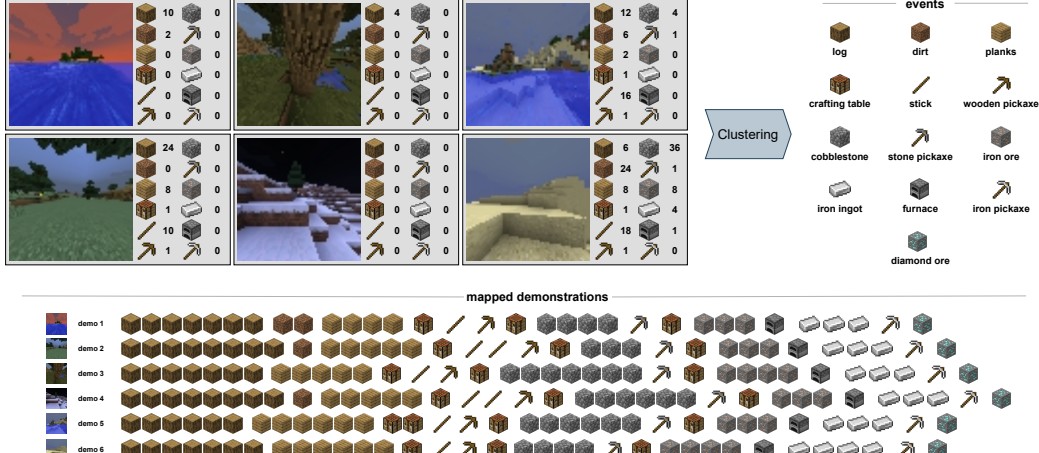

Figure A.11: Step (**I**): Define events and map demonstrations into sequences of events. First, we extract the sequence of states from human demonstrations, transform images into feature vectors using a pre-trained network and transform them into a sequence of consecutive state deltas (concatenating image feature vectors and inventory states). We cluster the resulting state deltas and remove clusters with a large number of members and merge smaller clusters. In the case of demonstrations for the *ObtainDiamond* task in Minecraft the resulting clusters correspond to obtaining specific resources and items required to solve the task. Then we map the demonstrations to sequences of events.

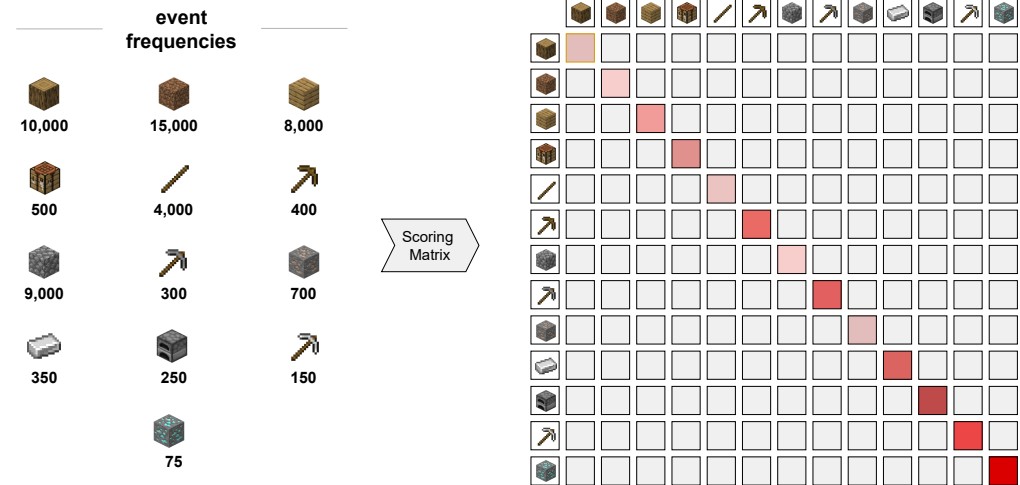

Figure A.12: Step (**II**): Construct a scoring matrix using event probabilities from demonstrations for diagonal elements and setting off-diagonal to a constant value. The scores in the diagonal position are proportional to the inverse of the event frequencies. Thus, aligning rare events has higher score. Darker colors signify higher score values.

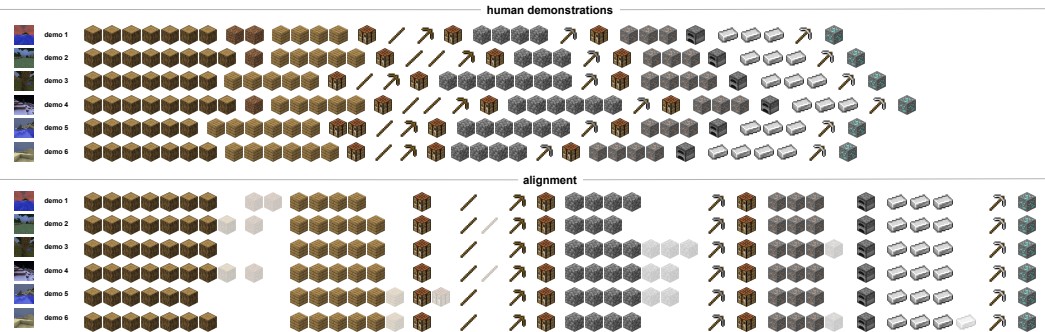

Figure A.13: Step (III) Perform multipe sequence alignment (MSA) of the demonstrations. The MSA algorithm maximizes the pairwise sum of scores of all alignments. The score of an alignment at each position is given by the scoring matrix. As the off-diagonal entries are negative, the algorithm will always try to align an event to itself, while giving preference to events which give higher scores.

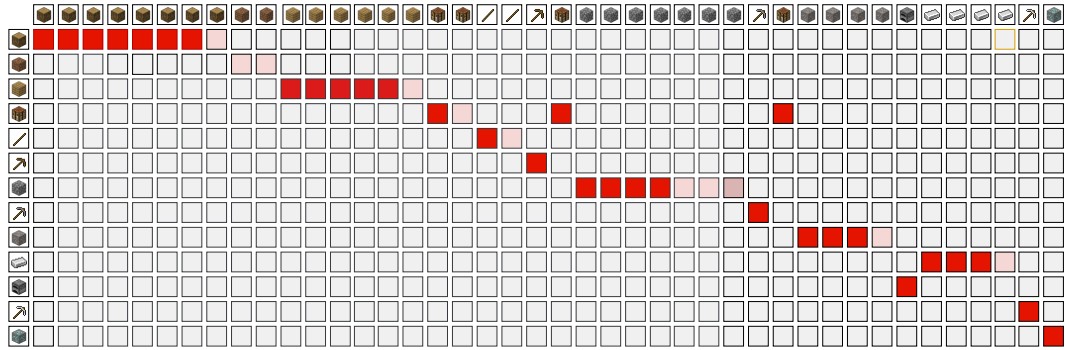

Figure A.14: Step (IV) Compute a position-specific scoring matrix (PSSM). This matrix can be computed using the MSA (Step (III)) and the scoring matrix (Step (II)). Every column entry is for a position from the MSA. The score at a position (column) and for an event (row) depends on the frequency of that event at that position in the MSA. For example, the event in the last position is present in all the sequences, and thus gets a high score at the last position. But it is absent in the remaining position, and thus gets a score of zero elsewhere.

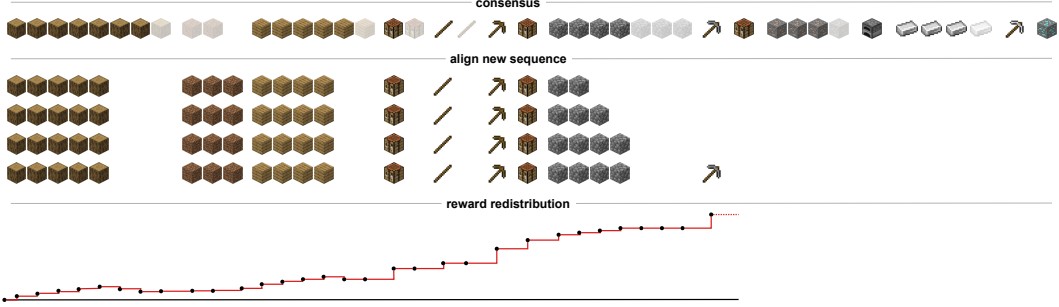

Figure A.15: Step (V) A new sequence is aligned step by step to the profile model using the PSSM, resulting in an alignment score for each sub-sequence. The redistributed reward is then proportional to the difference of scores of subsequent alignments.

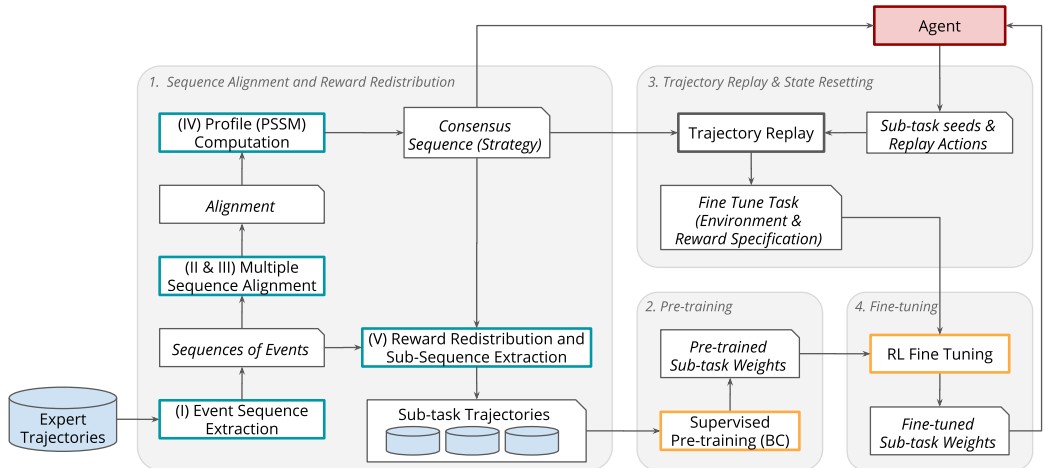

Figure A.16: Conceptual overview of our MineRL agent.

### A.7.4 Implementation of our Algorithm for Minecraft

The architecture of the training pipeline incorporates three learning stages:

- sequence alignment and reward redistribution
- learning from demonstrations via behavioral cloning (pre-training) and
- model fine-tuning with reinforcement learning.

Figure A.16 shows a conceptual overview of all components.

**Sequence alignment and reward redistribution.** First, we extract the sequence of states from human demonstrations, transform images into feature vectors using a standard pre-trained network and transform them into a sequence of consecutive state deltas (concatenating image feature vectors and inventory states). A pre-trained network can be model trained for image classification or an auto-encoder model trained on images. In our case, we used an auto-encoder model trained on the MineRL obtainDiamond dataset. We cluster the resulting state deltas and remove clusters with a large number of members and merged smaller clusters. This results in 19 events and we map the demonstrations to sequences of events. These events correspond to inventory changes. For each human demonstration we get a sequence of events which we map to letters from the amino acid code, resulting in a sequence of letters. In Fig. A.19 we show all events with their assigned letter encoding that we defined for the Minecraft environment.

We then calculate a scoring matrix according to step (II) in Sec. 3 in the main document. Then, we perform multiple sequence alignment to align sequences of events of the top $N$ demonstrations, where shorter demonstrations are ranked higher. This results in a sequence of common events which we denote as the consensus. In order to redistribute the reward, we use the PSSM model and assign the respective reward. Reward redistribution allows the sub-goal definition i.e. positions where the reward redistribution is larger than a threshold or positions where the reward redistribution has a certain value. In our implementation sub-goals are obtained by applying a threshold to the reward redistribution. The main agent is initialized by executing sub-agents according to the alignment. Figure 4 shows how sub-goals are identified using reward redistribution.

**Learning from demonstrations via behavioral cloning.** We extract demonstrations for each individual sub-task in the form of sub-sequences taken from all demonstrations. For each sub-task we train an individual sub-agent via behavioral cloning.

**Model fine-tuning with reinforcement learning.** We fine-tune the agent in the environment using PPO (Schulman et al., 2018). During fine-tuning with PPO, an agent receives reward if it manages to reach its sub-goal.

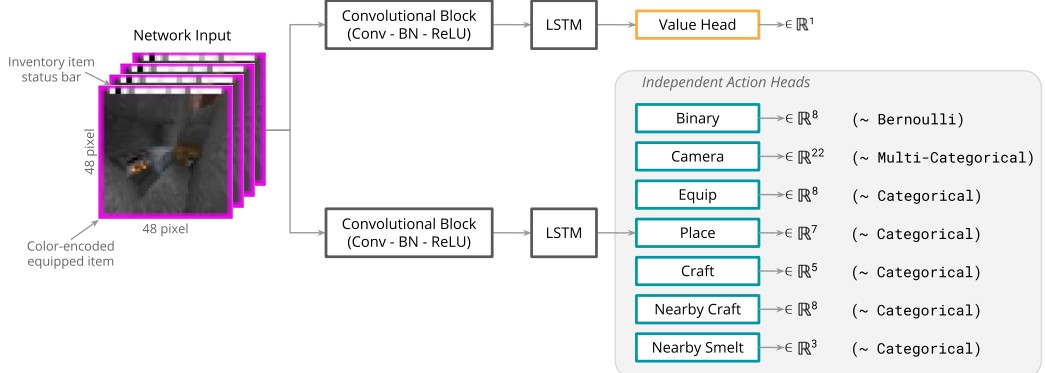

Figure A.17: Conceptual architecture of our MineRL policy and value networks.

To evaluate the performance of an agent for its current sub-goal, we average the return over multiple roll-outs. This gives us a good estimate of the success rate and if trained models have improved during fine tuning or not. In Fig. 6, we plot the overall success rate of all models evaluated sequentially from start to end.

In order to shorten the training time of our agent, we use trajectory replay and state resetting, similar to the idea proposed in (Ecoffet et al., 2019), allowing us to train sub-task agents in parallel. This is not necessary for the behavioral cloning stage, since here we can independently train agents according to the extracted sub-goals. However, fine-tuning a sub-task agent with reinforcement learning requires agents for all previous sub-tasks. To fine-tune agents for all sub-tasks, we record successful experiences (states, actions, rewards) for earlier goals and use them to reset the environment where a subsequent agent can start its training. In Fig. A.20, we illustrate a trajectory replay given by an exemplary consensus.

### A.7.5   POLICY AND VALUE NETWORK ARCHITECTURE

Figure A.17 shows a conceptual overview of the policy and value networks used in our MineRL experiments. The networks are structured as two separate convolutional encoders with an LSTM layer before the respective output layer, without sharing any model parameters.

The input to the model is the sequence of the 32 most recent frames, which are pre-processed in the following way: first, we add the currently equipped item as a color-coded border around each RGB frame. Next, the frames are augmented with an inventory status bar representing all 18 available inventory items (each inventory item is drawn as an item-square consisting of $3 \times 3$ pixels to the frame). Depending on the item count the respective square is drawn with a linearly interpolated gray-scale ranging from white (no item at all) to black (item count $> 95$). The count of 95 is the 75-quantile of the total amount of collected cobblestones and dirt derived from the inventory of all expert trajectories. Intuitively, this count should be related to the current depth (level) where an agent currently is or at least has been throughout the episode. In the last pre-processing step the frames are resized from $64 \times 64$ to $48 \times 48$ pixels and divided by 255 resulting in an input value range between zero and one.

The first network stage consists of four batch-normalized convolution layers with ReLU activation functions. The layers are structured as follows: Conv-Layer-1 (16 feature maps, kernel size 4, stride 2, zero padding 1), Conv-Layer-2 (32 feature maps, kernel size 4, stride 2, zero padding 1), Conv-Layer-3 (64 feature maps, kernel size 3, stride 2), and Conv-Layer-4 (32 feature maps, kernel size 3, stride 2). The flattened latent representation ($\in \mathbb{R}^{32 \times 288}$) of the convolution stage is further processed with an LSTM layer with 256 units. Given this recurrent representation we only keep the last time step (e.g. the prediction for the most recent frame).

The value head is a single linear layer without non-linearity predicting the state-value for the most recent state. For action prediction, two types of output heads are used depending on the underlying action distribution. The binary action head represents the actions *attack*, *back*, *forward*, *jump*, *left*, *right*, *sneak* and *sprint* which can be executed concurrently and are therefore modeled based on a

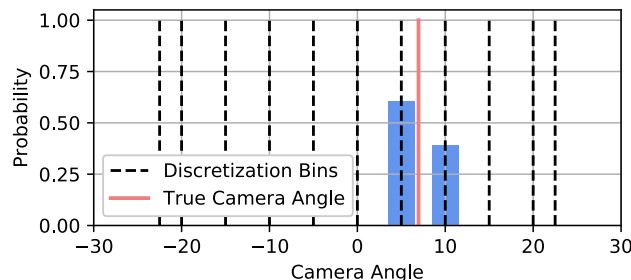

Figure A.18: Discretization and interpolation of camera angles.

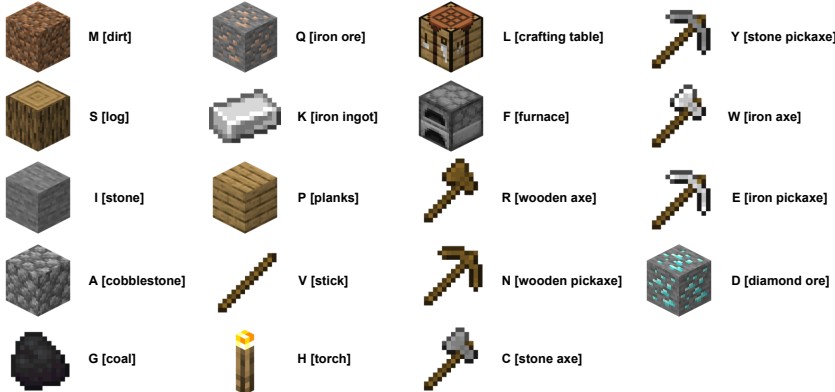

Figure A.19: Mapping of clusters to letters.

*Bernoulli* distribution. Since only one item can be equipped, placed, or crafted at a time these actions are modeled with a *categorical* distribution. The equip head selects from *none*, *air*, *wooden-axe*, *wooden-pickaxe*, *stone-axe*, *stone-pickaxe*, *iron-axe* and *iron-pickaxe*. The place head selects from *none*, *dirt*, *stone*, *cobblestone*, *crafting-table*, *furnace* and *torch*. The craft head selects from *none*, *torch*, *stick*, *planks* and *crafting-table*. Items which have to be crafted nearby are *none*, *wooden-axe*, *wooden-pickaxe*, *stone-axe*, *stone-pickaxe*, *iron-axe*, *iron-pickaxe* and *furnace*. Finally, items which are smelted nearby are *none*, *iron-ingot* and *coal*. For predicting the camera angles (*up/down* as well as *left/right*) we introduce a custom action space outlined in Figure A.18. This space discretizes the possible camera angles into 11 distinct bins for both orientations leading to the 22 output neurons of the camera action head. Each bin holds the probability for sampling the corresponding angle as a camera action, since in most of the cases the true camera angle lies in between two such bins. We share the bin selection probability by linear interpolation with respect to the distance of the neighboring bin centers to the true camera angle. This way we are able to train the model with standard categorical cross-entropy during behavioral cloning and sample actions from this categorical distribution during exploration and agent deployment.

### A.7.6 IMITATION LEARNING OF SUB-TASK AGENTS

Given the sub-sequences of expert data separated by task and the network architectures described above we perform imitation learning via behavioral cloning (BC) on the expert demonstrations. All sub-task policy networks are trained with a cross-entropy loss on the respective action distributions using stochastic gradient decent with a learning rate of $0.01$ and a momentum of $0.9$. Mini-batches of size 256 are sampled uniformly from the set of sub-task sequences. As we have the MineRL simulator available during training we are able to include all sub-sequences in the training set and evaluate the performance of the model by deploying it in the environment every 10 training epochs. Once training over 300 epochs is complete we select the model checkpoint based on the total count of collected target items over 12 evaluation trials per checkpoint. Due to presence of only successful sequences, the separate value network is not pre-trained with BC.

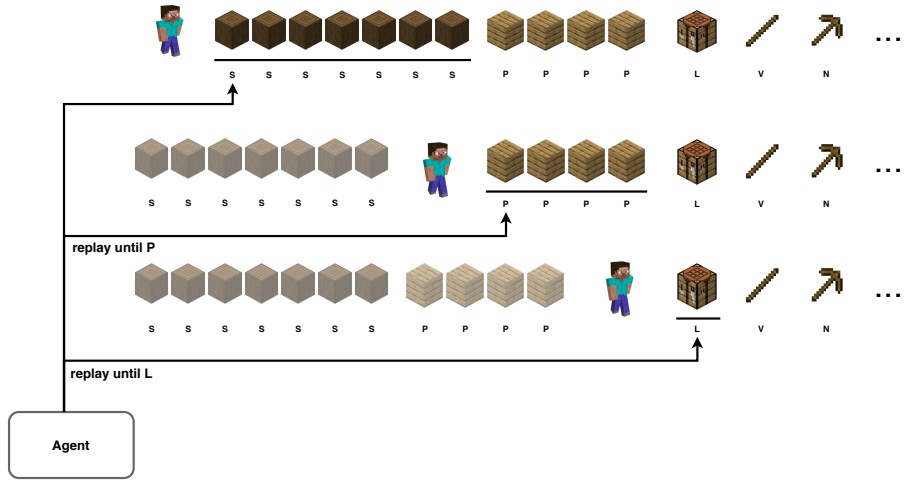

Figure A.20: Trajectory replay given by an exemplary consensus. The agent can execute training or evaluation processes of various sub-tasks by randomly sampling and replaying previously recorded trajectories on environment reset. Each letter defines a task. L (log), P (planks), V (stick), L (crafting table) and N (wooden pickaxe).

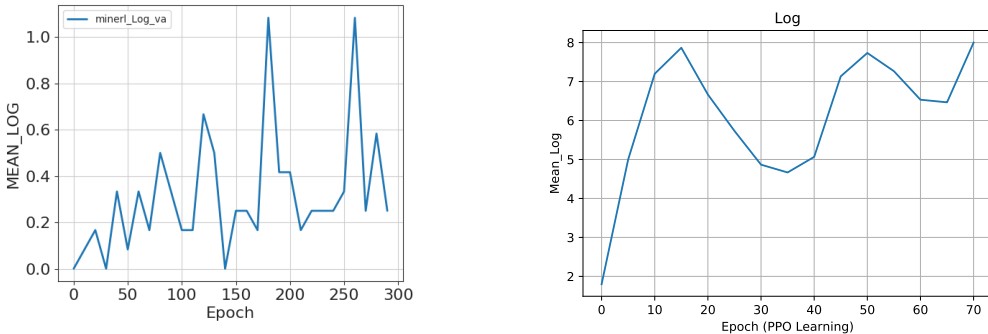

Figure A.21: Average number of logs collected during training: left: Behavioral Cloning, Right: PPO Training

### A.7.7 REINFORCEMENT LEARNING ON SUB-TASK AGENTS

After the pretraining of the Sub-Task agents, we further fine tune the agents using PPO in the MineRL environment. The reward is the redistributed reward given by Align-RUDDER. The value function is initialized in a burn-in stage prior to policy improvement where the agent interacts with the environment for 50k timesteps and only updates the value function. Finally, both policy and the value function are trained jointly for all sub-tasks. All agents are trained between 50k timesteps and 500k timesteps. We evaluate each agent periodically during training and in the end select the best performing agent per sub-task. A.21 - A.25 present evaluation curves of some sub-task agents during learning from demonstrations using behavioral cloning and learning online using PPO.

### A.8 REPRODUCING THE ARTIFICIAL TASK RESULTS

The code to reproduce the results and figures of both artificial tasks is provided as supplementary material. The README contains step-by-step instructions to set up an environment and run the experiments. By default, instead of using 100 seeds per experiment only 10 are used in the demonstration code.

Finally, a video showcasing the MineCraft agent is also provided as supplementary material.

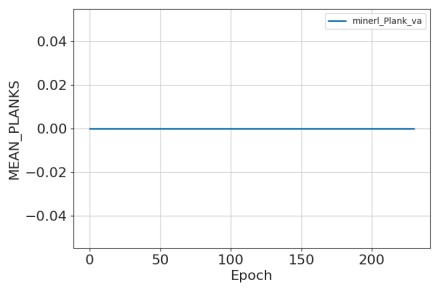 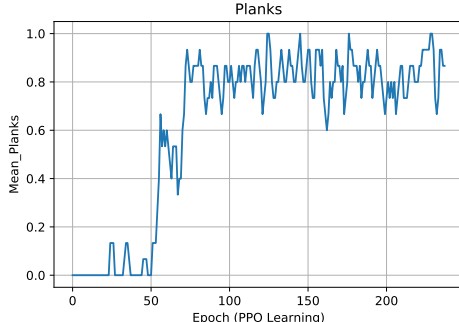

Figure A.22: Average number of planks crafted during training: left: Behavioral Cloning, Right: PPO Training

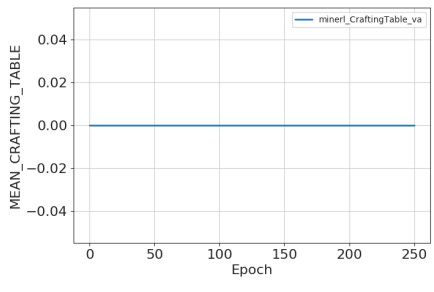 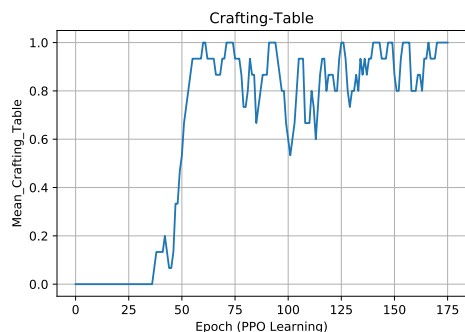

Figure A.23: Average number of table crafted during training: left: Behavioral Cloning, Right: PPO Training

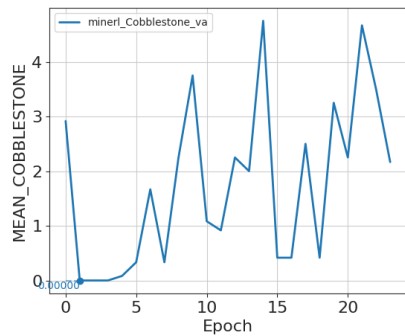 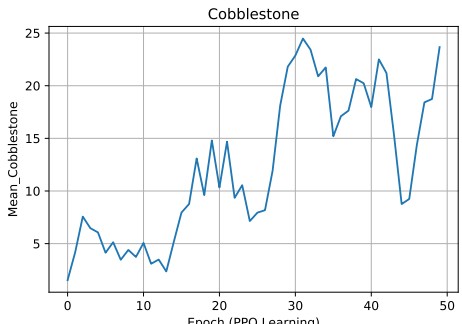

Figure A.24: Average number of stone collected during training: left: Behavioral Cloning, Right: PPO Training

## A.9 SOFTWARE LIBRARIES

We are thankful towards the developers of Mazelab Zuo (2018), PyTorch Paszke et al. (2019), OpenAI Gym Brockman et al. (2016), Numpy Harris et al. (2020), Matplotlib Hunter (2007) and Minecraft Guss et al. (2019b).

## A.10 COMPUTE

Artificial task (I) and (II) experiments were performed using CPU only as GPU speed-up was negligible. The final results for all methods were created on an internal CPU cluster with 128 CPU

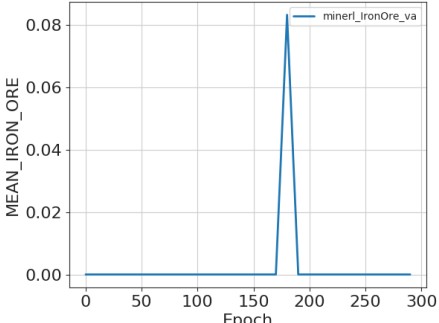 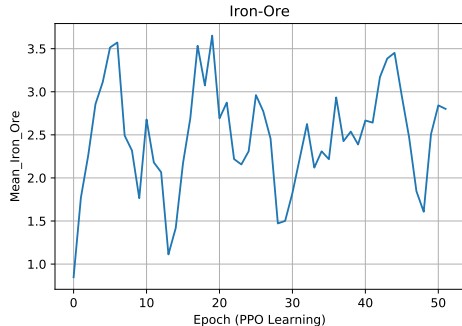

Figure A.25: Average number of iron-ore collected during training: left: Behavioral Cloning, Right: PPO Training

cores with a measured wall-clock time of 10,360 hours. The majority of compute is spent on baseline methods.

For minecraft, during development 6 to 8 nodes each with 4 GPUs of an internal GPU cluster were used for roughly six months of GPU compute time (Nvidia Titan V and 2080 TI).

The compute required for training the final agent was well within the challenge parameters (4 days on a single node with one GPU).

