# OpenReview forum: "Align-RUDDER: Learning From Few Demonstrations by Reward Redistribution"
_ICLR.cc/2022/Conference — ICLR 2022 Submitted_

### Official Review · Reviewer_YcqX · 2021-10-30

**Correctness:** 4
**Technical Novelty And Significance:** 2
**Empirical Novelty And Significance:** 3
**Recommendation:** 6
**Confidence:** 3

**Main Review:**

## **Paper Strengths**

**Motivation**: The authors make very clear the motivation. Improving RUDDER, a reward redistribution method, so that it can work on tasks in which demonstrations of high-reward trajectories are given because the task is too hard for exploration alone.

**Method Novelty:** Applying a sequence alignment mechanism seems relevant and a novel contribution, also leading to good empirical results.

**Experiment Results:** Align-RUDDER seems to have great performance improvements over RUDDER, especially with a very limited number of demonstrations. The compared baselines also seem to be relevant, demonstrating Align-RUDDER's good performance. Furthermore, the MineCraft diamond mining demonstrates impressive performance gains. The authors also perform hypothesis testing, further validating the results of their experiments.

**Supplementary Material:** The authors provide a detailed set of background knowledge for RUDDER, sequence alignment, and extra figures/examples in the appendix. Furthermore, the supplementary video was nice and informative.



## **Paper Weaknesses**

**Grammatical Issues:** A few grammatical issues throughout, it doesn't detract too much but occasionally causes a hiccup in the flow when reading, so please fix these.

**Method Clarity**:

- The paper figures do not explain the method too well. Figure 2 is a decent figure, but the caption does not really explain the right half, nor does it draw any parallel between the two parts of the figures. I think this figure should be simplified, labelled, and used to better show how the reward redistribution using conservation score mechanism works in the biological sequence alignment case, and in the reward redistribution case. Figure 3 is nice, but it should have some text explaining things better in the caption. As the actual explanation for how the method + alignment strategy works is very complicated, these figures are critical for showing it clearly at a higher level.
- Furthermore, there should be some more intuition presented earlier about *why* sequence alignment is easier than RUDDER's LSTM.
- In Section 3, "Reward Redistribution by Sequence Alignment," to someone unfamiliar with biological alignment techniques, you should explain why alignment will intuitively help first, before diving into details.
- In fact, in general, I think at least to me the amount of detail given to the alignment explanation should be redistributed for a conference like ICLR. The authors should rewrite the method section to have far fewer details (except those explicitly needed) about the alignment algorithm itself, this should all be in the appendix. Instead, the entire reward redistribution scheme should be used to explain both intuitively and in detail how this 1) encourages alignment of similar trajectories in terms of states and actions, 2) better alignment scores  result in a better reward redistribution scheme, and 3) is far easier to learn with few demonstrations than RUDDER's LSTM. Of course, some details should be kept but in general I think there is too much detail placed in the nitty-gritty of *how* alignment is done in the main paper.

**Experiments:** To me, the experimental setup seems valid. But along the same lines as what was stated above, there should be a bit more analysis in the main text about *why* Align-RUDDER performs the way it does in comparison to the other methods.

## **Questions**

- Appendix page 33: "We...transform images into feature vectors using a standard pre-trained network," what standard pre-trained network was used?

## Minor Issues

- Contribution 2 and 4 in the intro are saying basically the same thing
- "Sub-tasks" in page 5, is this meant to have a new line before and be bolded instead?
- Page 7, describing the hierarchical setup, "more details can be found in the appendix," please link an appendix section for easier referencing here
- The reference to learning methods should be a clickable link back to that subsection describing the learning method (i had forgotten the learning methods by the time I reached page 6 where it's mentioned in the experiments)


**Summary Of The Paper:**

This paper presents an improvement over RUDDER to allow for RUDDER-style reward redistribution when given limited sets of demonstrations by using a biological sequence alignment model.


**Summary Of The Review:**

In summary, I think this paper presents strong results, and is a valid technical contribution, but the main reason I am currently voting for reject is that it needs some rewriting to address my clarity concerns. I am willing to adjust my score if the authors can improve on this.

---

> ### Author Response · Authors · 2021-11-19
> **Response to reviewer Ycqx**
>
> We thank the reviewer for pointing out both the strengths and weaknesses of our work.
> The arguments helped us to improve our paper and we believe it is much more approachable and understandable now. We were also quite pleased to know that you found our paper to be a “relevant and novel contribution” with “strong results”.
>
> **Regarding figures:**
>
> Thanks for pointing this out, we expanded the caption of Figure 2 to better show the similarity between conservation score and reward redistribution. We will further improve Figure 2 for the final version of the paper. We also explain the five steps in the caption of Figure 3 and also improved captions for other figures.
>
> **Regarding “why” sequence alignment is better than LSTM:**
>
> Thank you for suggesting this. To explain better why alignment is better than LSTM with few expert demonstrations, we include two paragraphs. One explaining the limitations of LSTM and another paragraph on how alignment overcomes these limitations. We also discuss the advantages of alignment further.
>
> We also performed additional experiments in a controlled toy environment without optimizing a policy to show that sequence alignment is much better at identifying key-events than the LSTM model used in RUDDER, especially when only few demonstrations are available.
>
> To that end, we use a simple 1D environment where the agent has to collect a key and then open a chest, receiving its reward at the end of the fixed length episode. Since the key-events in this simple environment are known we can compute the success of each method in assigning credit to these key-events. Especially with a small number of demonstrations, sequence alignment significantly outperforms the LSTM models.
>
> In these experiments, Align-RUDDER was able to detect ~96% of the events while LSTM models detected only ~46% (averaged over different numbers of training demonstrations, 1000 test episodes and 10 seeds). Please see the paragraph titled “Alignment vs LSTM in 1D key-chest environment”, in the experiments section.
>
> **Regarding writing:**
>
> We have made the writing much more simpler and attempt to give more intuition now. In particular, we improved the presentation of the five steps from defining events to redistributing reward. We have made it more simple and moved the extra details to the appendix.
>
> **Regarding pre-trained network:**
>
> A pre-trained network can be model trained for image classification or an auto-encoder model trained on images. In our case, we used an auto-encoder model trained on the MineRL obtainDiamond dataset. We clarify this in the appendix.
>
> **Regarding Minor Issues:**
>
> Finally, we addressed the minor issues mentioned by the reviewer and continue to improve the appendix for the camera ready version.

---

> > ### Comment · Reviewer_YcqX · 2021-11-21
> > **Response to authors**
> >
> > Thanks for the response.
> >
> > **Re: New experiments**
> > This is a nice experiment, thanks for including this.
> >
> > **Re: writing/figure/clarity improvements**
> > Overall, I think the writing is a bit improved, and Figure 2 has been greatly improved to clarify the intuition about sequence alignment.
> > However, there are still issues with the writing as a whole, and it can still be improved.
> >
> > For example, the paragraph starting with "Reward Redistribution: Idea and return decomposition", is hard to read as nearly every sentence re-references Figure 1. Figure 1 should be completely self-contained so that this paragraph does not need to reference it in every sentence; it's a bit jarring to read.
> >
> > Another example: "The profile model is the result of a multiple sequence alignment of the demonstrations and allows aligning new sequences to it" What does "aligning new sequences to it" mean in this context? It's not until later that I realized the meaning of this sentence.
> >
> > Furthermore, the five step method section is now improved for sure, but I think it can still be further improved written more clearly.
> >
> > In summary, because of the improved clarity and experiments I will be raising my score. But in the current state, I think the paper is *almost there* in terms of being good enough for ICLR, mainly due to writing. As such i will be raising from a 3 to a 5.

---

> > > ### Author Response · Authors · 2021-11-22
> > > **Paper updated again, please have a look**
> > >
> > > Thank you very much for actively helping us to improve our paper. We fully agree that writing should not be of any hindrance for publication. Following your latest feedback, we have updated the paper draft including the paragraphs you mentioned.
> > >
> > > Concretely:
> > > * We have improved the paragraph explaining reward redistribution
> > > * We have further improved the caption of Figure 1, it is now self-contained
> > > * We have further improved Section 3, such that we first give an intuitive introduction into how Align-RUDDER works before going into details. We believe that this has much improved the readability of this section, and makes it easier to follow our method.
> > > * We have further improved the caption of Figure 3 such that we explain the five steps of Align-RUDDER in more detail.
> > > * We have also improved the five step method section, please have a look
> > > * We still found grammar errors which we corrected
> > >
> > > Furthermore, we have introduced a new Appendix (section A.7.3) where we put additional emphasis on the explanation of the 5 steps of Align-RUDDER based on the example of Minecraft. To support our explanation, we have created 5 new figures (Figures A.11 to A.15), and walk the interested reader through this appendix.
> > >
> > > We again thank the reviewer for the detailed help to improve the paper and hope that we have resolved the remaining issues. In our opinion, this iteration has helped a lot to improve the paper. If accepted, we will keep improving the writing of our work.

---

> > > > ### Comment · Reviewer_YcqX · 2021-11-27
> > > > **Response to authors again**
> > > >
> > > > Thanks for the paper revisions.
> > > >
> > > > While there are still quite a few obvious grammatical errors, I believe that my main concerns are essentially addressed. As such, i have again raised my score to a 6.

---

### Official Review · Reviewer_nk2L · 2021-11-01

**Correctness:** 2
**Technical Novelty And Significance:** 3
**Empirical Novelty And Significance:** 3
**Recommendation:** 3
**Confidence:** 4

**Main Review:**

While the results in Minecraft are presented as impressive, the paper contains several shortcomings from the strict scientific point of view. The key question - is the new method working better than the original RUDDER and why? - is poorly addressed. The comparison is made only on two different GridWorld problems, and it is unclear whether the benefit is from the new reward redistribution technique or division to subtasks. An ablation study and experiments on different domains, perhaps those used in the original RUDDER paper, are required to find the answer.

Moreover, some key concepts of the paper are poorly explained. Section "Defining Events" does not explain how the clustering is exactly done. A function "f(s, a)" remain undefined. I don't agree to the authors' claim that "Q-function of an optimal policy resembles a step function" and "is mostly constant". Presentation of the paper can be improved: It is not clear why the biological sequences in Fig. 2 left is included. Fig. 6 include letters on x-axis without any explanation about their meaning. Grammar and typography can be improved.


**Summary Of The Paper:**

The paper uses Sequence Alignment technique to redistribute rewards, to a similar effect as with LSTM in RUDDER. The hierarchical agent is trained with behavioral cloning and fine-tuned with RL (tabular in Rooms environment / PPO in MineRL). Tasks are automatically divided into subtasks and specialized agents used for the subtasks. The main contributions seem to be: a) introduction of the Sequence Alignment technique which works well with few expert demonstrations; b) experimental demonstration in Rooms / MineRL.

**Summary Of The Review:**

Given all the drawbacks, the paper is not ready for publication. I would recommend the authors to focus more on the in-depth comparison with Rudder, save some space by moving the figures regarding Minecraft to the appendix, rewrite the key concepts in more depth, verify the key assumptions and demonstrate WHY the new method is better.

---

> ### Author Response · Authors · 2021-11-19
> **Response to reviewer nk2L**
>
> Thank you for your review, your suggestions helped us to improve the paper tremendously. We were also quite pleased to know that you found our minecraft results “impressive”.
>
> **Regarding LSTM (RUDDER) vs Sequence Alignment (Align-RUDDER):**
>
> We concur that we neglected to address the advantages of Align-RUDDER over the original RUDDER sufficiently. Therefore, we performed additional experiments in a controlled toy environment without optimizing a policy to show that sequence alignment is much better at identifying key-events than the LSTM model used in RUDDER, especially when only few demonstrations are available. In these experiments, Align-RUDDER was able to detect ~96% of the events while LSTM detected only ~46% (averaged over different training demonstrations, 1000 test episodes and 10 seeds). Please see the paragraph titled “Alignment vs LSTM in 1D key-chest environment”, in the experiments section.
>
> **Regarding “why” alignment is better than LSTM with few demonstrations:**
>
> To explain better why alignment is better than LSTM with few expert demonstrations, we have now included a paragraph on the limitations of the LSTM with few demonstrations. Further, we have added another paragraph on how alignment overcomes these limitations and also discuss the advantage of alignment.
>
> **Regarding sub-tasks in GridWorld experiments:**
>
> We apologize for not being clear. For the GridWorld experiments we did not divide the problem into sub-tasks. In the earlier version of the paper, we did mention dividing the $Q$-table to smaller sub-tables for sub-tasks. We did this to establish a connection between reward redistribution and sub-tasks. We have improved the description of the GridWorld experiments to clarify that in these experiments the problem is not divided into sub-tasks explicitly, therefore showing that the reward redistribution is the reason for the performance improvements over the other methods.
>
> **Regarding the “five steps”:**
>
> We improved the writing, giving more details where necessary and more intuitive explanations as other reviewers also pointed out deficiencies in this section.
>
> **Regarding $Q$-function being a step function:**
>
> The reviewer is correct in pointing out that our statement regarding the $Q$-function resembling a step-function is not general. It only applies to complex tasks that are hierarchically composed of sub-tasks. We now state this more clearly. Thank you for pointing this out.
>
> **Regarding writing:**
>
> Finally, we attempt to improve the overall presentation by, among others, improving figure captions, including a better explanation of the similarities between biological sequences and strategies in Figure 2 and a reference to the mapping of letters to events in Figure 6.

---

> > ### Comment · Reviewer_nk2L · 2021-11-27
> > **Revision**
> >
> > Dear authors, thank you for the revision.
> >
> > While it addresses some of the concerns and improves the paper, I still think that a major revision is required. I would start with the overall narrative, and would not emphasize the Minecraft environment that much. I would like to see much more detailed comparison to the RUDDER algorithm. I would add experiments with the 4 Atari domains used in the RUDDER paper for comparison. The Figure showing how the original reward and redistributed reward looks like would be especially nice (see for example Figure 2 in the RUDDER paper for inspiration).
> >
> > Given that, I am keeping the original score. But I encourage the authors to revise the paper properly and resubmit.

---

### Official Review · Reviewer_mK3T · 2021-11-04

**Correctness:** 3
**Technical Novelty And Significance:** 3
**Empirical Novelty And Significance:** 3
**Recommendation:** 5
**Confidence:** 4

**Main Review:**

Strengths:

- Simple, effective and general technique for identifying subgoals with few expert demonstrations sharing a common strategy

- Large gains over RUDDER and sota imitation learning methods on 2 gridworld tasks

- The only technique that was able to mine a diamond in the minecraft challenge with automatically learned subgoals (infrequently)

Current Limitations:

- As shown in figure 6, the success rate of Align-RUDDER is high, but then degrades rapidly in the last quarter of the minecraft task. An analysis of what elements of Align-RUDDER contributed to that degradation would be highly instructive in understanding the current limitations of the approach, and future research directions.

- Align-RUDDER's "five steps" are described completely enough, but I feel that the presentation needs to be further improved for those not already familiar with profile models (many). Expanding figure 3 and better connecting the description with it would improve the paper substantially.

- The assumption of a single underling strategy in Align-RUDDER seems like a significant limitation. A clustering step to identify multi-strategy profiles will be required in many situations. How common such situations are and how they would be detected and mitigated with e.g. with multi-strategy profiles is not adequately discussed.
- Figure 1, which describes the basic intuitions behind the RUDDER approach, distributes all of the reward to earlier subtasks, leaving none for the end-goal, which seems like an issue...

Post-rebuttal comments:

Thank you to the authors for their response. I have looked at the other reviews and re-read parts of the updated paper, and I tend to agree with the more critical reviews wrt the following:

1) The explanation of RUDDER is still not complete or detailed enough to appreciate Align-RUDDER's relative strengths (and weaknesses), and thus the contributions of the paper.

2) The explanation of Align-RUDDER needs significant work wrt both the layering and quality of explanation, it is just not clear enough. Read the first sentence in (V) of section 3, does it make sense? There are many more examples of poorly crafted sentences. Perhaps more significantly, it just doesn't come together to explain the technique clearly. In my opinion it still needs to be completely revised.

3) In addition, following up on my concern with figure 1, I also take issue with the "Q-functions are step-functions statement", and the characterization around figure 1 that RUDDER makes all EFRs zero... These are not the correct characterizations. RUDDER is representing the EFR with a (sparse) set of reward advantage events through reward redistribution, and as such, the details around RUDDER become very important--perhaps with better "event detection" regularization, the LSTM approach is far more effective than the Align approach, which is constructed in a somewhat ad-hoc manner with a diverse set of tools.

Based on these considerations, the paper is still in need of substantial revision, and I have lowered my score.

**Summary Of The Paper:**

The authors propose Align-RUDDER, which aligns given task demonstrations (sharing a common strategy) using a profile model, to identify common events as subgoals and redistibute reward to realize more efficient and effective RL. Align-RUDDER replaces RUDDER's replay buffer with task demonstrations and RUDDER's LSTM with a simpler profile model to maximally leverage high reward but scarce demonstrations in tasks characterized by sparse and delayed reward signals, which are notoriously difficult to explore effectively.

**Summary Of The Review:**

A solid approach to the difficult and important problem of learning from scarce expert demonstrations. An analysis of the success rate degradation in the last part of the minecraft task, and an improved explanation of Align-RUDDER's "five steps" would strengthen the paper significantly.

---

> ### Author Response · Authors · 2021-11-19
> **Response to reviewer Mk3t**
>
> Thank you for a very elaborate and profound review and suggestions that helped us to improve our paper. We were also quite pleased to know that you found our method to be “simple, effective and general technique”.
>
>
> **Regarding Align-RUDDER’s success rate in MineCraft:**
>
> Thank you for your suggestion. Now we include a section analyzing our MineCraft results, which also includes an explanation why the success rate decreases. During learning from demonstrations, much less training data is available for later subtasks. Not all expert demonstrations achieve the later tasks. During online training using reinforcement learning, an agent has to successfully complete all earlier subtasks to generate trajectories for later sub-tasks. This is exponentially difficult. The lack of demonstrations and difficulty of the agent to generate data for later sub-tasks leads to degradation of the success rate in MineCraft.
>
> **Regarding Align-RUDDER “five steps”:**
>
> We have improved the writing for Align-RUDDER’s five steps and now give a more intuitive description in the main paper. We moved the details to the appendix. We also changed the caption of Figure 3 to include an explanation of the five steps.
>
> **Regarding multiple strategies:**
>
> Multiple sequence alignment already performs clustering (over sequences) and obtains a tree internally with multiple strategies. The most prominent strategy will then dominate the alignment. In our experiments, we have only considered single strategy problems. Extending Align-RUDDER to switch between multiple strategies is part of ongoing research. Though, it should not be too difficult to achieve. For multiple strategies, Align-RUDDER would include an additional clustering step over sequences before alignment. In this manner, only sequences which are clustered together will be aligned. This permits using Align-Rudder also for multiple strategies.
>
> **Regarding Figure 1:**
>
> In this environment, if the agent gets the key and opens the door, the agent will always receive the treasure at the end of the episode. This is needed to add artificial delay between key-actions and the reward. Therefore, the redistribution in the right panel is correct. Such delay exists in numerous real world problems and makes learning difficult. We improved the explanation of this scenario to make this clearer now.

---

> ### Author Response · Authors · 2021-11-22
> **Response to updated comments**
>
> Thank you for updating your comments. We have updated the paper draft and improved the writing.
>
> Concretely:
> * We have improved the paragraph explaining reward redistribution
> * We have further improved the caption of Figure 1, it is now self-contained
> * We have further improved Section 3, such that we first give an intuitive introduction into how Align-RUDDER works before going into * details. We believe that this has much improved the readability of this section, and makes it easier to follow our method.
> * We have further improved the caption of Figure 3 such that we explain the five steps of Align-RUDDER in more detail.
> * We have also improved the five step method section, please have a look
> * We still found grammar errors which we corrected
>
> **New explanation for 5 steps on Minecraft:**
> Furthermore, we have introduced a new Appendix (section A.7.3) where we put additional emphasis on the explanation of the 5 steps of Align-RUDDER based on the example of Minecraft. To support our explanation, we have created 5 new figures (Figures A.11 to A.15), and walk the interested reader through this appendix.
>
> **Regarding expected future reward (EFR):**
> We have to disagree with the reviewer regarding this. RUDDER does try to make the expected future reward zero. See Theorem 2 in RUDDER[1]. By doing this, RUDDER reduces the delay in reward and makes learning faster.
>
> [1] RUDDER: Return Decomposition for Delayed Rewards, Arjona-Medina et.al, 2019
>
> **Regarding $Q$-function being a step function:**
> The reviewer is correct in pointing out that our statement regarding the $Q$-function resembling a step-function is not general. It only applies to complex tasks that are hierarchically composed of sub-tasks. We now state this more clearly. Thank you for pointing this out.
>
> **Regarding the explanation of RUDDER**:
> We have improved the explanation of RUDDER and made it more simple. We also include a more detailed explanation of RUDDER in the appendix.
>
> We again thank the reviewer for the detailed help to improve the paper and hope that we have resolved the writing issues.

---

### Official Review · Reviewer_bJpP · 2021-11-05

**Correctness:** 3
**Technical Novelty And Significance:** 2
**Empirical Novelty And Significance:** 2
**Recommendation:** 6
**Confidence:** 4

**Details Of Ethics Concerns:**

Nil

**Main Review:**

Strengths:
+ The empirical evaluation is well laid out and the experiments are described with necessary details. The authors perform careful evaluation on multiple (synthetic) environments to build intuition for and motivate the changes in the training algorithm.

Weakness:
+ The paper writing could be improved to better explain the prior work and segregating the core contributions of the work (e.g reward redistribution builds on RUDDER).
+ The experiments are restricted to navigation based grid-world style environments. It would be useful to have more comparison on other benchmark tasks like locomotion.

**Summary Of The Paper:**

The paper considers the challenge of improving sample efficiency of RUDDER-style algorithms in sparse MDPs. Building on prior work by Arjona-Medina et al [1], the authors incorporate demonstrations of optimal trajectories from an expert in the training pipeline. Additionally, to improve the sample efficiency and stability, the authors replace LSTM-model of RUDDER with an alignment based profile model.

The approach is evaluated on two synthetic grid-world based environments and a MineCraft based environment. On both benchmarks the proposed algorithm works better than baseline RUDDER.

**Summary Of The Review:**

The authors consider the challenge of improving sample efficient HRL in sparse environments. They identify several drawbacks in RUDDER and propose effective modifications to the learning algorithm to improve performance. While the experiments are promising they are restricted to navigation style experiments, it would be useful to have comparison to more imitation learning based baselines with diversity in learning environments.

---

> ### Author Response · Authors · 2021-11-19
> **Response to reviewer bJpP**
>
> Thank you for your review and suggestions. It has helped us to improve our paper. We were also pleased to know that you found our “empirical evaluation well laid out”.
>
> **Regarding writing:**
>
> We have simplified the writing a lot after suggestions from you and other reviewers. We have attempted to segregate text regarding prior work (RUDDER) and our core contributions (Align-RUDDER). The entire prior work is now in the section “Review of RUDDER”. We also have added a paragraph on the limitations of LSTM to further support our motivations to use alignment for reward redistributions.
>
> **Regarding experiments:**
>
> We have included a new experiment to *support* why Align-RUDDER performs so well compared to RUDDER’s LSTM. We show sequence alignment can detect key events much better than a LSTM, when there are very few demonstrations available. Please see the paragraph titled “Alignment vs LSTM in 1D key-chest environment” in the experiment section.
>
> **Regarding locomotion experiments:**
>
> Thank you for your suggestion. We are currently performing experiments with more environments including locomotion tasks (Mujoco) and will include the results in the appendix for the final version of the paper.

---

> > ### Comment · Reviewer_bJpP · 2021-11-30
> > **Response to authors**
> >
> > Thank you for the revised version. While the readability of the draft has improved, the presentation of the paper leaves scope for further improvement. Additionally, the current version of the paper focuses primarily on Minecraft tasks. To assess the general performance of the algorithm, it would be useful to have included experiments on other standard RL benchmarks with sparse rewards.
> >
> > With this context, I'd keep my original score.

---

### Author Response · Authors · 2021-11-19
**Response to reviewers**

We want to thank all reviewers for their time and valuable suggestions.

We were pleased to see that reviewers found the paper to be a “relevant and novel contribution” and contains  “impressive” results for Minecraft. We were also elated to find that some reviewers found Align-RUDDER as a “simple, effective and general technique” with “good empirical results”.

Please have a look at the updated version of the paper. We have included your suggestions and this has improved the paper. We added an additional experiment showing the advantages of sequence alignment over LSTM models for reward redistribution. Furthermore, we improved the presentation of our method in general and especially of the five steps of Align-RUDDER. We added a discussion regarding the advantages of sequence alignment for reward redistribution, specifically in the regime of few demonstrations. Finally, we addressed all minor issues and will continue to correct grammatical issues for the camera ready version.

We again would like to thank all reviewers for their time, insights and the overall high quality of the reviews.

---

### Decision · Program_Chairs · 2022-01-20

**Decision:**

Reject

**Comment:**

## A Brief Summary
This paper proposes two critical modifications to the original RUDDER algorithm:
1. Proposes the Align-RUDDER method that assumes that the episodes with high rewards can be used as demonstrations.
2. Uses a profile model from the Multiple sequence alignment approach to align the demonstrations and redistribute the rewards according to how frequently events in the demos are shared across different demonstrators. MSA is being used as a profile model instead of LSTM.

The paper uses successor features to represent state-action pairs, which is then used to compute the similarity matrix used for MSA afterward.
The paper shows promising results in the Minecraft environment (ObtainDiamond task,) as well as synthetic grid-world environments.

## Reviewer bJbP
*Strengths:*
- Empirical evaluation is well-done.
*Weaknesses:*
- The writing requires more work.
- Limited experiments: Mostly on toy-grid world/navigation environments, it is not clear if the results will generalize to the control problems.

## Reviewer mK3T
*Strengths:*
- Simple and effective technique for identifying sub-goals.
- Large improvements over original RUDDER.
- Impressive results on Minecraft.
*Weaknesses:*
- More through ablations on the importance of different elements of Align-RUDDER.
- Presentation and writing need more improvements.
- Assumption of a single underlying successful strategy is an important limitation.
- Figure 1 is problematic and confusing because of the way it explains the RUDDER algorithm.

## Reviewer nk2L
*Strengths:*
- Impressive results on Minecraft.
*Weaknesses:*
- Poor justification and motivation.
- RUDDER vs Align-RUDDER comparisons are only done on two grid-world environments.
- More ablations are required to justify the approach.
- Writing requires more work, some important concepts require more clarity. Some undefined concepts...
- Incorrect claims such as:
> Q-function of an optimal policy resembles a step function

## Reviewer YcqX
*Strengths:*
- Strong motivation.
- MSA for demos is novel.
- Strong experimental results.
*Weaknesses:*
- Several grammatical errors.
- The method is not explained well in the paper, the writing needs more work to improve the clarity.
- Lack of sufficient analysis and ablations on the Align-RUDDER approach.

## Key Takeaways and Thoughts
Overall, the result provided in this paper in the Minecraft environment is impressive. The motivation for the Align-RUDDER is clear for me. I like the paper; in particular, the application of the MSA for the alignments across the demos is novel. However, as all the reviewers of this paper agreed that the paper is unclear, especially the method description requires more work. The paper needs to present more ablations and analysis to justify which components of Align-RUDDER algorithm are essential. I agree with both insights, the authors have made improvements in the paper to improve the exposition of the algorithms, but still, the paper feels a bit rushed. I would recommend the authors reconsider the paper's current structure and improve the writing further, especially the description of the method can be further improved. I would recommend that the authors fix those essential issues with the paper and the other comments reviewers made in a future resubmission.